# Repurposing Synthetic Data for Fine-grained Search Agent Supervision

**Yida Zhao**[1,2,3]**, Kuan Li**[3]**, Xixi Wu**[3]**, Liwen Zhang**[3]***Dingchu Zhang**[3]**,**
**Baixuan Li**[3]**, Maojia Song**[3]**, Zhuo Chen**[1,2,3]**, Chenxi Wang**[3]**, Xinyu Wang**[3]**,**
**Kewei Tu**[1,2]***Pengjun Xie**[3]**, Fei Huang**[3]**, Jingren Zhou**[3]**, Yong Jiang**[3]*
[1] School of Information Science and Technology, ShanghaiTech University
[2] Shanghai Engineering Research Center of Intelligent Vision and Imaging
[3] Tongyi Lab, Alibaba Group

## Abstract

LLM-based search agents are increasingly trained on entity-centric synthetic data to solve complex, knowledge-intensive tasks. However, prevailing training methods like Group Relative Policy Optimization (GRPO) discard this rich entity information, relying instead on sparse, outcome-based rewards. This critical limitation renders them unable to distinguish informative "near-miss" samples—those with substantially correct reasoning but a flawed final answer—from complete failures, thus discarding valuable learning signals. We address this by leveraging the very entities discarded during training. Our empirical analysis reveals a strong positive correlation between the number of ground-truth entities identified during an agent's reasoning process and final answer accuracy. Building on this insight, we introduce **E**ntity-aware **G**roup **R**elative **P**olicy **O**ptimization (**E-GRPO**), a novel framework that formulates a dense entity-aware reward function. E-GRPO assigns partial rewards to incorrect samples proportional to their entity match rate, enabling the model to effectively learn from these "near-misses". Experiments on diverse question-answering (QA) and deep research benchmarks show that E-GRPO consistently and significantly outperforms the GRPO baseline. Furthermore, our analysis reveals that E-GRPO not only achieves superior accuracy but also induces more efficient reasoning policies that require fewer tool calls, demonstrating a more effective and sample-efficient approach to aligning search agents.

## 1 Introduction

The advent of Large Language Models (LLMs) has catalyzed the development of sophisticated autonomous agents, with **search agents** emerging as a prominent class for solving complex, knowledge-intensive tasks (Yao et al., 2023; Wang et al., 2024; Xi et al., 2025). Training these agents to navigate the vast, noisy web effectively requires abundant and challenging data (Google Team, 2025b; OpenAI, 2025; xAI Team, 2025; Moonshot AI, 2025). To meet this demand, a dominant paradigm of synthetic data generation has emerged (Wu et al., 2025b; Li et al., 2025b; Gao et al., 2025). In this paradigm, as shown in Figure 1 (left), complex questions are often created by systematically transforming simple "seed" questions through operations like fact injection or deliberate obfuscation. This process creates an intricate problem structure, paved with key entities that form the factual backbone of the correct answer.

This synthetic data is then used to train agents within the now-dominant reinforcement learning (Wen et al., 2024; Singh et al., 2025), especially with Group Relative Policy Optimization (GRPO) (Shao et al., 2024) and its numerous variants (Yu et al., 2025; Dong et al., 2025; Hu, 2025; Xu et al., 2024; Zhao et al., 2025). These methods typically rely on outcome-based rewards, utilizing only the final answer while discarding the intermediate entity information meticulously embedded during data synthesis. This mechanism leads to the reward sparsity problem (Qian et al., 2025; Deng et al., 2025), which manifests critically for search agents (Song et al., 2025; Wu et al., 2025a; Jin et al., 2025; Li et al., 2025c; Zheng et al., 2025; Zhang et al., 2025a; Li et al., 2025b; Gao

*Corresponding authors. tukw@shanghaitech.edu.cn, {zlw439616, yongjiang.jy}@alibaba-inc.com

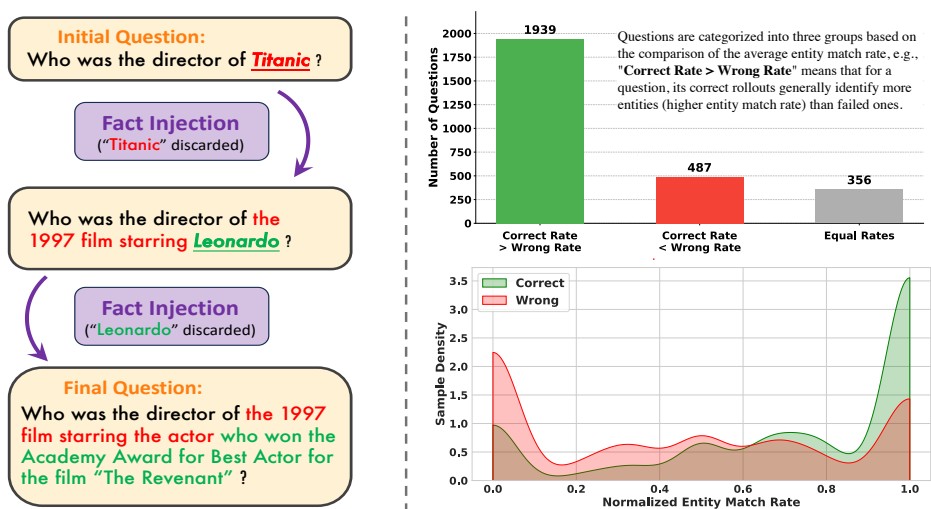

Figure 1: **Left:** An Example of entity-centric synthetic data generation. **Right:** Analysis of the correlation between entity matching and agent performance.

et al., 2025): by treating all negative samples uniformly, GRPO fails to distinguish a "near-miss"—a response with correct intermediate reasoning steps but a flawed answer—from a complete failure. For instance, in answering *Who was the director of the 1997 film starring the actor who won the Academy Award for Best Actor for the film 'The Revenant'?*, a "near-miss" that correctly identifies the actor (*Leonardo*) and the film (*Titanic*) but fails on the final answer is far more informative than one that misunderstands the query entirely. By penalizing both equally, standard GRPO discards crucial learning signals embedded in partially correct reasoning, forcing the model to re-learn steps it had already mastered.

One natural approach to address this sparse reward problem is to incorporate fine-grained, process-level supervision. In domains such as mathematics and code, this is achieved either by evaluating each intermediate step with a Process Reward Model (PRM) (Fan et al., 2025; Anonymous, 2025; Zhang et al., 2025b) or by employing complex sampling mechanisms (e.g., tree-based search) to derive step-level advantages (Yang et al., 2025; Hou et al., 2025). However, these methods are ill-suited for the open-ended nature of web search. The sheer scale and dynamic nature of web content render the annotation required for a PRM prohibitively expensive. Similarly, the extensive length of search agent trajectories, often involving dozens of tool calls and reasoning steps, makes intricate sampling strategies computationally intractable.

This leaves a critical gap: how can we obtain a fine-grained, informative, yet computationally efficient reward signal for search agents? The answer, we find, lies hidden in plain sight: within the very **entity-centric** information from synthetic data generation that GRPO-like methods discard. These entities, forming the factual backbone for the answer, intuitively represent an untapped source of fine-grained supervision. To validate the potential of these ground-truth entities, we analyze the relationship between agent performance and the number of entities matched during reasoning (**entity match rate**). As illustrated in Figure 1 (right), the strong positive correlation we observed (further discussed in Section 3.1) validates our core hypothesis: the entity match rate serves as a powerful proxy for factual correctness and can be repurposed as a fine-grained reward signal that standard GRPO lacks.

Based on this core insight, we propose **E**ntity-aware **G**roup **R**elative **P**olicy **O**ptimization (**E-GRPO**), a novel RL framework that enhances policy optimization by formulating a dense, entity-aware reward function from the entities within the synthetic training data. Specifically, instead of applying a uniform penalty, our method assigns a bonus to negative samples proportional to their entity match rate. By doing so, a "near-miss" sample, which contains many correct entities and is highly informative for learning, receives a better reward than a complete failure. This fine-grained reward, obtained with negligible computational cost, compels the model to move beyond simply avoiding errors and towards mastering the process of identifying and synthesizing key information, thereby addressing the limitation of standard GRPO in complex search tasks.

Our comprehensive evaluation on 11 benchmarks, spanning diverse models and environments, demonstrates that E-GRPO significantly and consistently surpasses the GRPO baseline. Critically, beyond superior accuracy, E-GRPO also enables more efficient reasoning policies that consistently require fewer tool calls. Further analyses validate our core hypothesis, confirming the importance of the entity-aware reward.

In summary, the key contributions of this work are as follows:

- We identify the "near-miss" problem in GRPO-based training and propose the core insight that entities from synthetic data can be repurposed as a fine-grained reward signal, supported by empirical analysis revealing a strong correlation between entity match rate and task accuracy.

- We introduce **E-GRPO**, a novel framework that enhances policy optimization by formulating an entity-aware reward function to differentiate the quality of negative samples and provide more granular supervision.

- We conduct experiments on multiple QA and deep research benchmarks, demonstrating that E-GRPO not only outperforms the GRPO baseline in accuracy but also learns more efficient policies.

## 2 PRELIMINARY

In this section, we provide a brief overview of key concepts in search agents and a review of entity-centric data synthesis methods. More discussion of related work is available in Section 5.

### 2.1 SEARCH AGENTS

**Multi-turn Rollout.** We adopt the ReAct (Yao et al., 2023) paradigm for search agents. The LLM agent iteratively performs `thought` and `action`, and receives `observation` from the environment. Specifically, in each iteration, the LLM agent generates a free-form `thought` ($\tau$) and executes a valid `action` (e.g., a tool call $a$). Then it waits for the environment's feedback as the `observation` ($o$). In the web search setting, the `action` space typically consists of searching queries, webpage browsing, and generating the final answer. The iteration terminates when the LLM generates a final answer. A complete rollout with $T$ iterations can be defined as:

$$\mathcal{H} = \big(\tau_1, a_1, o_1, \ldots, \tau_t, a_t, o_t, \ldots, \tau_T, a_T\big),$$

where $\tau_t, a_t, o_t$ represent thought, action and observation at step $t$, with $\tau_t, a_t$ sampled from a policy $\pi_\theta$ based on all previous context as $(\tau_t, a_t) \sim \pi_\theta(\cdot \mid q, \tau_1, a_1, o_1, \ldots, \tau_{t-1}, a_{t-1}, o_{t-1})$. The specific format of multi-turn rollout is detailed in Appendix A.

**Tool Design.** Following existing search agent studies (Li et al., 2025b; Gao et al., 2025), we define the agent's web exploration `action` space with two essential tools:

- **Search**: A search engine that accepts multiple queries and retrieves the top-10 relevant results per query, including titles, snippets, and the corresponding URLs.

- **Visit**: A browser agent that accesses several web pages simultaneously, given the corresponding URLs and browsing goals. It first retrieves the full webpage and then uses Qwen3-30B-A3B-Instruct-2507 (Team, 2025a) to extract relevant information based on the browsing goal.

### 2.2 ENTITY-CENTRIC DATA SYNTHESIS

A significant line of research has focused on the autonomous generation of complex and grounded question-answer (QA) pairs (Wu et al., 2025a;b; Li et al., 2025b; Gao et al., 2025), sharing a common thread in their entity-centric approach. We briefly summarize two state-of-the-art (SOTA) methods that exemplify this paradigm below.

- **ASearcher** (Gao et al., 2025): Starting with a seed question, ASearcher's synthesis agent iteratively increases difficulty via two entity-focused operations: Injection, which replaces named entities with descriptive facts, and Fuzzing, which substitutes specific entities with more ambiguous, general descriptions.

- **SailorFog-QA** (Li et al., 2025b): SailorFog-QA first constructs a complex knowledge graph via a random walk from a seed entity, creating intricate entity couplings. It then generates questions by sampling subgraphs and applying information obfuscation, which involves replacing specific entity attributes with vague descriptions.

## 3 METHODOLOGY

In this section, we first give a detailed analysis of the correlation between agent performance and synthetic-data entity matching. Then, we propose the E-GRPO algorithm, designed to improve GRPO with a fine-grained entity-aware reward function.

### 3.1 ANALYZING ENTITY MATCHING IN AGENTIC REASONING

Inspired by the entity-centric approach for data generation, where entities are intuitively the factual backbone of the synthetic data, we conduct an empirical analysis to investigate how these entities correlate with the performance of a search agent.

**Metrics.** To quantify this correlation, we first define the **entity match rate**. Given a synthetic QA pair $(q, gt)$, we retain all the $m$ ground-truth entities during QA generation $E_q = \left\{ e^{(1)}, e^{(2)}, \ldots, e^{(m)} \right\}$, and sample a group of $G$ rollouts $\left\{ \mathcal{H}^{(1)}, \mathcal{H}^{(2)}, \ldots, \mathcal{H}^{(G)} \right\}$. For each rollout $\mathcal{H}^{(i)}$ in the group, let $\mathcal{T}^{(i)} = \left\{ \tau_1^{(i)}, \tau_2^{(i)}, \ldots, \tau_{T_i}^{(i)} \right\}$ be the collection of all thoughts in rollout $i$. We identify the set of entities matched within the thoughts as:

$$E_{\text{matched}}^{(i)} = \left\{ e \in E_q \mid \exists\, t \in \{1, \ldots, T_i\}, e \text{ is mentioned in } \tau_t^{(i)} \right\}, \tag{1}$$

An entity is considered "mentioned" if its full phrase appears as an exact string match in the thought's text (more discussion available in Appendix B). The **entity match rate** for rollout $i$, denoted as $\gamma_i$, is then calculated as the ratio of matched entities to the total:

$$\gamma_i = \frac{\left| E_{\text{matched}}^{(i)} \right|}{|E_q|} = \frac{\left| E_{\text{matched}}^{(i)} \right|}{m} \tag{2}$$

Furthermore, to enable robust comparison across different questions which may have varying difficulty, we introduce the **normalized entity match rate**, $\hat{\gamma}_i$. This is calculated by normalizing the raw rate $\gamma_i$ against the maximum rate, $\gamma_{\max}$, observed within its question group:

$$\hat{\gamma}_i = \begin{cases} \dfrac{\gamma_i}{\gamma_{\max}} & \text{if } \gamma_{\max} > 0 \\ 0 & \text{otherwise} \end{cases} \quad \text{where} \quad \gamma_{\max} = \max_{j \in \{1, \ldots, G\}} \gamma_j. \tag{3}$$

This normalization allows us to aggregate the match rate of all rollouts on a common 0-to-1 scale.

**Analysis.** To investigate the correlation between **entity match rate** and accuracy, we first conduct a per-question analysis on a sampled subset of SailorFog-QA (Li et al., 2025b) using the WebSailor-7B agent (Li et al., 2025b). For each question, we perform 8 rollouts and calculate the average entity match rates of correctly solved and failed rollouts, respectively (further explanation in Appendix I). As shown in Figure 1 (upper right), the average entity match rate of correct rollouts was higher than that of failed ones in the vast majority of the questions, outnumbering the reverse scenario by a clear 4-to-1 margin (1939 vs. 487 questions). This establishes a strong correlation between the entity match rate and the correctness of the final answer.

Moving beyond this aggregate, per-question view, we analyze the distribution of the **normalized entity match rate** across all individual rollouts. As shown in Figure 1 (bottom right), the distributions for correct and incorrect rollouts diverge significantly. The distribution of correct samples (green) peaks sharply at a normalized rate of 1.0. In contrast, incorrect samples (red) show a bimodal distribution: a large peak at 0.0, and a notable spread across the mid-to-high range. This

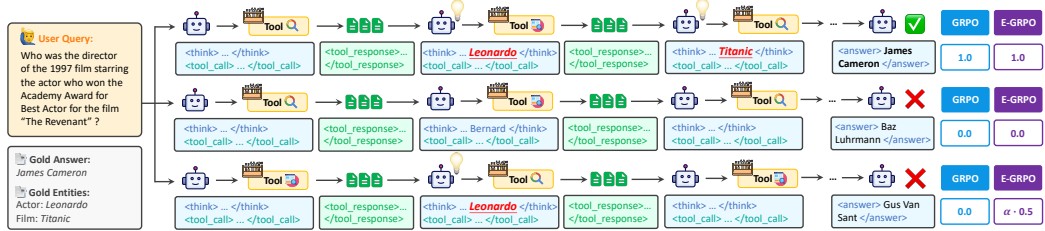

Figure 2: Comparison of GRPO and E-GRPO. GRPO applies outcome-based reward, while E-GRPO additionally assigns a bonus to negatives proportional to their **normalized entity match rate**. The three rollouts illustrate a success, a complete failure, and a "near-miss", respectively.

latter group represents the informative "near-misses", where most entities were found but the final reasoning failed.

This analysis shows that the entity match rate is more than just a pass/fail indicator. Instead, it provides a granular scale to distinguish valuable "near-misses" from complete failures, offering a richer signal of an agent's reasoning quality.

## 3.2 ENTITY-AWARE GROUP RELATIVE POLICY OPTIMIZATION

The preceding analysis shows that entity match rate offers a fine-grained signal of an agent's reasoning quality. Conventional policy optimization methods, however, largely ignore this signal by relying on a sparse, outcome-based reward tied only to answer correctness, thereby treating all failures as equally undesirable. Therefore, we introduce **Entity-aware Group Relative Policy Optimization (E-GRPO)**, a framework that directly incorporates the entity match rate into its reward function to guide policy learning better.

**Limitations of Reward Formulation in GRPO.** Existing GRPO-like frameworks (Shao et al., 2024) for search agents typically employ outcome-based reward. Specifically, the reward $R_i$ for a rollout $\mathcal{H}^{(i)}$ is defined simply as 1 if it leads to a correct answer, and 0 otherwise. This reward is then used to compute a group-relative advantage. This advantage value, denoted as $\hat{A}_{i,j}$, is calculated once for the entire rollout $i$ and then applied to every token $j$ within it, serving as the core learning signal:

$$\hat{A}_{i,j} = \frac{R_i - \text{mean}(\{R_k\}_{k=1}^G)}{\text{std}(\{R_k\}_{k=1}^G)}. \tag{4}$$

The limitation of this formulation is evident: as shown in Figure 2, standard GRPO assigns an identical reward of 0 to both a complete failure (middle rollout, 0 entities matched) and an informative "near-miss" (bottom rollout, 1 entity matched), thus rendering their different reasoning qualities indistinguishable.

**Entity-aware Reward Formulation.** E-GRPO addresses the limitation of outcome-based rewards by redefining the reward function with an entity-aware bonus. We utilize the **normalized entity match rate** $\hat{\gamma}_i$ rather than the raw rate, as its consistent 0-to-1 scale is essential for a stable advantage calculation across different groups. Our entity-aware reward is thus defined as:

$$R_i = \begin{cases} 1 & \text{if } \mathcal{H}^{(i)} \text{ is correct} \\ \alpha \cdot \hat{\gamma}_i & \text{if } \mathcal{H}^{(i)} \text{ is wrong} \\ 0 & \text{if error[1] occurs in } \mathcal{H}^{(i)} \end{cases}, \tag{5}$$

where $\alpha \in [0, 1]$ is a hyperparameter balancing the value of accuracy and entity matching. This formulation yields two significant advantages. (1) It creates a dense reward spectrum to distinguish the quality of incorrect rollouts. As shown in Figure 2, a "near-miss" that identifies a correct entity (*Leonardo*) is rewarded ($\alpha \cdot 0.5$), unlike a complete failure which receives zero. (2) It provides a meaningful training signal even in all-wrong groups where standard GRPO offers no gradient.

---

[1]Errors (format and overlength problems) are detailed in the subsequent paragraph **Implementation Details**.

**Overall Training Objective.** With our entity-aware reward defined, we can now formalize the complete E-GRPO objective. First, the refined reward from Eq. 5 is used to compute a more informative advantage $\hat{A}_{i,j}$ via Eq. 4. The policy is then optimized by maximizing the GRPO objective $\mathcal{J}(\theta)$, defined as:

$$
\mathcal{J}(\theta) = \mathbb{E}_{(q, gt) \sim \mathcal{D}, \{\mathcal{H}^{(i)}\}_{i=1}^{G} \sim \pi_{\theta_{\text{old}}}}
$$
$$
\left[ \frac{1}{\sum_{i=1}^{G} |\mathcal{H}^{(i)}|} \sum_{i=1}^{G} \sum_{j=1}^{|\mathcal{H}^{(i)}|} \min\left( r_{i,j}(\theta)\hat{A}_{i,j}, \text{clip}(r_{i,j}(\theta), 1 - \varepsilon_{\text{low}}, 1 + \varepsilon_{\text{high}})\hat{A}_{i,j} \right) \right], \quad (6)
$$

where $r_{i,j}(\theta) = \frac{\pi_\theta(\mathcal{H}_j^{(i)}|q, \mathcal{H}_{j-1}^{(i)})}{\pi_{\theta_{\text{old}}}(\mathcal{H}_j^{(i)}|q, \mathcal{H}_{j-1}^{(i)})}$ is the importance sampling ratio.

**Implementation Details.** Based on this objective, we additionally apply the following practical modifications to the training of all models (both our method and the baselines):

- **KL-Free Objective and Policy Exploration.** Following DAPO (Yu et al., 2025), we remove the KL-divergence regularization term in GRPO and apply the "clip-higher" method, which increases the upper clipping bound $\varepsilon_{\text{high}}$, to better encourage policy exploration.

- **Handling Format Errors.** Rollouts with format errors (defined in Appendix A) are assigned a reward of 0. This strict penalty is justified because our RL training is preceded by a cold-start SFT phase that ensures the model is already familiar with the required output format.

- **Handling Overlength Rollouts.** Overlength rollouts (i.e., those exceeding token or tool-call limits) are also assigned a reward of 0. We observed in preliminary experiments that directly optimizing on these rollouts can lead to policy collapse. Therefore, we adopt a specific handling strategy: while these rollouts contribute to the advantage normalization (i.e., computing the group's mean and standard deviation), they are excluded from the final loss computation to prevent instability.

## 4 EXPERIMENTS

### 4.1 EXPERIMENT SETUP

**Benchmarks.** Our evaluation spans a diverse set of 11 benchmarks to comprehensively assess E-GRPO's effectiveness. For question-answering tasks, we use three single-hop datasets: Natural Questions (NQ) (Kwiatkowski et al., 2019), TriviaQA (TQ) (Joshi et al., 2017), and PopQA (Mallen et al., 2022); and four multi-hop datasets: 2WikiMultiHopQA (2Wiki.) (Ho et al., 2020), HotpotQA (HQA) (Yang et al., 2018), Bamboogle (Bamb.) (Press et al., 2022), and MuSiQue (Musi.) (Trivedi et al., 2022). We further test our agent on four challenging deep research benchmarks: GAIA (Mialon et al., 2023), BrowseComp (Wei et al., 2025), BrowseComp-ZH (Zhou et al., 2025), and xbench-DeepSearch (xbench-DS) (Xbench-Team, 2025). Following Asearcher (Gao et al., 2025), we use 1000 sampled instances from the validation sets of HQA, 2Wiki., and Musi. For GAIA, we use the 103-sample text-only validation subset (Li et al., 2025c). For all other benchmarks, we use their full test sets.

**Baselines and Reference Agents.** Our primary baseline is the direct counterpart trained with GRPO (Shao et al., 2024), allowing for a controlled comparison of the algorithmic enhancement. We also compare against a suite of ReAct-based agents. For QA benchmarks, this includes R1-Searcher-7B (Song et al., 2025), DeepResearcher-7B (Zheng et al., 2025), Search-R1-32B (Jin et al., 2025), Simple-DS-QwQ (Sun et al., 2025), and ASearcher-14B (Gao et al., 2025). For deep research benchmarks, we include advanced models like OpenAI-o3, Claude-4-Sonnet (Anthropic, 2025), Kimi-K2 (Team et al., 2025), and DeepSeek-V3.1 (Liu et al., 2024), alongside open-source models with no more than 32B parameters such as R1-Searcher-7B, WebThinker-RL (Li et al., 2025d), WebDancer-QwQ (Wu et al., 2025a), and WebSailor-7B/32B (Li et al., 2025b).

**Environment Settings.** We conduct training in two distinct environments to validate E-GRPO's robustness: a closed-world **local knowledge base (Local)** and an open-world **web exploration**

Table 1: Overall **Pass@1** performance on standard QA benchmarks. Results with ‡ are sourced from Gao et al. (2025). The top scores of each evaluation environment are **bolded**.

| Environment | Model | Multi-Hop QA | | | | Single-Hop QA | | | Avg |
|---|---|---|---|---|---|---|---|---|---|
| | | 2Wiki. | HQA | Bamb. | Musi. | NQ | TQ | PopQA | |
| *Comparison among Our Models* | | | | | | | | | |
| Local | Local-7B-SFT | 74.0 | 66.7 | 72.8 | 30.2 | 49.8 | 78.4 | 49.6 | 60.2 |
| | Local-7B-GRPO | 75.1 | 65.1 | 74.4 | 31.2 | 51.5 | 82.0 | **50.4** | 61.4 |
| | Local-7B-E-GRPO | **79.6** | **69.0** | **78.4** | **32.8** | **55.8** | **83.9** | 50.2 | **64.2** |
| *Comparison with Other Reference Agents* | | | | | | | | | |
| Web | R1-Searcher-7B‡ | 69.4 | 61.6 | 72.0 | 25.3 | 48.7 | 79.5 | 45.2 | 57.4 |
| | DeepResearcher-7B‡ | 64.1 | 61.0 | 76.8 | 24.5 | 52.9 | 82.8 | 45.7 | 58.3 |
| | Search-R1-32B‡ | 69.3 | 64.2 | 81.6 | 30.8 | 51.1 | 86.6 | 53.6 | 62.5 |
| | Simple-DS-QwQ‡ | **80.4** | 67.5 | 83.2 | 32.9 | 55.3 | 90.2 | 47.8 | 65.3 |
| | ASearcher-14B‡ | 79.8 | 70.5 | 80.8 | 33.8 | 55.4 | 88.5 | **50.5** | 65.6 |
| | Local-7B-SFT | 76.8 | 70.7 | 80.2 | 32.2 | 55.4 | 88.7 | 48.9 | 64.7 |
| | Local-7B-GRPO | 77.2 | **73.8** | 82.4 | **34.9** | 55.9 | 89.3 | 50.1 | 66.2 |
| | Local-7B-E-GRPO | **80.4** | 73.7 | **85.6** | **34.9** | **59.1** | **90.4** | 50.2 | **67.8** |

**(Web)** environment. In the Local setting, search and visit tools are simulated via information retrieval over a Wikipedia 2024 corpus (Karpukhin et al., 2020; Gao et al., 2025). In the Web setting, the agent interacts with the live web using Google Search and Jina (Jina.ai, 2025) for page fetching.

**Training Details.** Our experiments are based on Qwen2.5-7B-Instruct (Yang et al., 2024) and Qwen3-30B-A3B-Instruct-2507 (Team, 2025a), covering different model sizes and architectures (dense and MoE). It is important to note that our study aims to **validate the effectiveness of E-GRPO at the algorithmic level**, not merely to pursue state-of-the-art performance. Therefore, we use limited data to ensure training efficiency while still enabling performance comparison.

- **Cold-start SFT:** We first fine-tune the base models on 11K samples from SailorFog-QA (Li et al., 2025b). This step, following Dong et al. (2025), mitigates reward collapse and ensures the model understands the agentic format before RL.

- **RL:** We generate two distinct 1k-sample datasets for RL training. For the Local environment, we synthesize data using the Asearcher (Gao et al., 2025) method over the 2024 Wikipedia corpus. For the Web environment, we use the SailorFog-QA data generation pipeline. Note that both methods are anchored in entities from Wikipedia despite the distinct environments they use. Crucially, for both datasets, we retain all ground-truth entities generated during the synthesis process to enable E-GRPO. We train the 7B model in both environments, while the 30B model is trained only in the more complex Web environment. For each setup, we apply both GRPO and E-GRPO.

We denote our models by their training environment, model sizes, and the training algorithm, e.g., **Local-7B-GRPO**. Detailed hyperparameters are presented in Appendix C.

**Evaluation Metrics.** Model answers, extracted from the model output enclosed in `<answer>` and `</answer>` tags (detailed in Appendix A), are evaluated for correctness using Qwen2.5-72B-Instruct under the LLM-as-Judge setting. We report the average **Pass@1** over all test samples, as well as the **Pass@3** across three rollouts.

## 4.2 MAIN RESULTS

We present the experiment results across three evaluation settings: (1) 7B models trained and evaluated with the Local environment on standard QA benchmarks, (2) the same 7B models evaluated with the Web environment on the same benchmarks, and (3) all models trained and evaluated with the Web environment on deep research benchmarks.

Table 2: Overall performance on four challenging deep research benchmarks. Results with † are sourced from Wu et al. (2025c). The top two Pass@1 scores of agents ≤ 32B are **bolded** and underlined. The top Pass@3 scores of our agents are **bolded**.

| Model | GAIA | | BrowseComp | | BrowseComp-ZH | | xbench-DS | |
|---|---|---|---|---|---|---|---|---|
| | Pass@1 | Pass@3 | Pass@1 | Pass@3 | Pass@1 | Pass@3 | Pass@1 | Pass@3 |
| *Advanced Models* | | | | | | | | |
| OpenAI-o3† | 70.5 | - | 50.9 | - | 58.1 | - | 66.7 | - |
| Claude-4-Sonnet† | 68.3 | - | 12.2 | - | 29.1 | - | 64.6 | - |
| Kimi-K2† | 57.7 | - | 14.1 | - | 28.8 | - | 50.0 | - |
| DeepSeek-V3.1† | 63.1 | - | 30.0 | - | 49.2 | - | 71.0 | - |
| *Open-source Agents ≤ 32B* | | | | | | | | |
| R1-Searcher-7B | 20.4 | - | 0.4 | - | 0.6 | - | 4.0 | - |
| WebThinker-RL | 48.5 | - | 2.8 | - | 7.3 | - | 24.0 | - |
| WebDancer-QwQ | 51.5 | - | 3.8 | - | 18.0 | - | 39.0 | - |
| WebSailor-7B | 37.9 | - | 6.7 | - | 14.2 | - | 34.3 | - |
| WebSailor-32B | **53.2** | - | 10.5 | - | 25.5 | - | **53.3** | - |
| *Our Agents* | | | | | | | | |
| Web-7B-SFT | 31.7 | 44.7 | 5.7 | 10.5 | 13.2 | 25.6 | 37.3 | 55.0 |
| Web-7B-GRPO | 33.0 | 44.7 | 6.3 | 11.7 | 17.5 | 31.5 | 40.7 | 56.0 |
| Web-7B-E-GRPO | 36.9 | 51.5 | 9.3 | 16.1 | 18.1 | 32.1 | 42.0 | 59.0 |
| Web-30B-SFT | 45.0 | 60.2 | 10.8 | 18.5 | 23.8 | 38.1 | 43.7 | 63.0 |
| Web-30B-GRPO | 47.6 | 62.1 | 12.3 | 18.9 | 25.7 | 38.8 | 45.3 | 65.0 |
| Web-30B-E-GRPO | 48.5 | **65.1** | **12.9** | **21.0** | **26.4** | **41.2** | 46.7 | **66.0** |

**Performance in the Local Environment on QA benchmarks.** The top block of Table 1 presents the results for models trained and evaluated within the controlled Local environment. Our Local-7B-E-GRPO model achieves the highest average score of 64.2, marking a substantial improvement of 2.8 points over the GRPO baseline and 4.0 points over the initial SFT model. This superiority is consistent across most individual benchmarks, demonstrating that the entity-aware reward allows the model to learn a more effective reasoning policy than the outcome-based reward.

**Performance in the Web Environment on QA benchmarks.** As shown in the second block of Table 1, even when evaluated with the unfamiliar web environment, our Local-7B-E-GRPO model again achieves the highest average score among its peers at 67.8, outperforming the GRPO counterpart and other open-source baselines with larger sizes. This result strongly validates the generalizability and robustness of our method, allowing a locally trained model to achieve superior performance in a completely different, real-world setting.

**Performance on Deep Research Benchmarks.** As presented in Table 2, results on deep research benchmarks consistently underscore the superiority of E-GRPO. Across both 7B and 30B scales, our E-GRPO models significantly outperform their GRPO counterparts. Notably, Web-30B-E-GRPO achieves the best performance among open-source agents on BrowseComp (12.9) and BrowseComp-ZH (26.4), even surpassing advanced models like Claude-4-Sonnet on BrowseComp, and narrows the gap with others.

The algorithmic advantage of E-GRPO is most evident in the Pass@3 results. While GRPO offers minimal gains over the SFT baseline (e.g., 44.7 on GAIA), E-GRPO delivers substantial improvements (e.g., a 6.8-point jump to 51.5). This stems from a key algorithmic difference: GRPO's outcome-based reward tends to refine existing successful strategies, whereas E-GRPO's entity-aware reward explicitly encourages exploring promising but incomplete paths. This allows the agent to build a more diverse set of solutions, which directly increases its chances of succeeding within a few attempts and explains the significant Pass@3 gains.

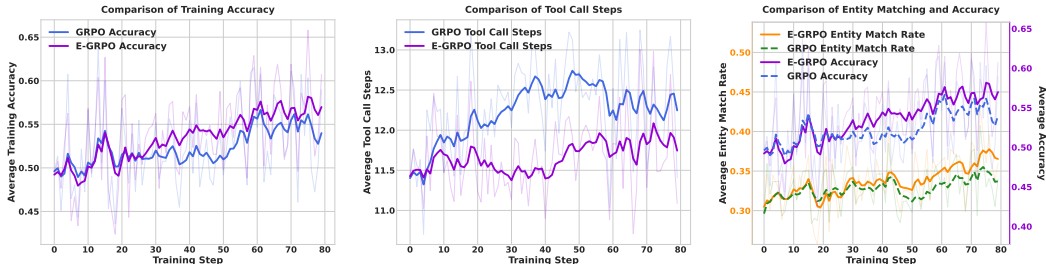

Figure 3: Training dynamics of 30B models with the Web environment, including the comparison of E-GRPO and GRPO over training accuracy, tool call steps, and the analysis between entity matching and training accuracy.

## 4.3 ANALYSIS

**Training Dynamics.** We begin by analyzing the training dynamics of E-GRPO against the GRPO baseline. As shown in Figure 3, E-GRPO demonstrates superior learning efficiency and effectiveness. The left panel shows that E-GRPO (purple) consistently achieves **higher training accuracy**, showing a steadier and more pronounced upward trend than the GRPO baseline (blue). This suggests that the dense, entity-aware reward provides a more effective and stable learning signal. Simultaneously, the middle panel reveals that E-GRPO learns a more efficient reasoning policy, consistently using **fewer tool calls** per rollout. This efficiency can be attributed to rewarding the discovery of key entities, which guides the agent towards more direct and informative solution steps. Extended training dynamics are provided in Appendix G for reference.

To further validate E-GRPO's mechanism, we analyze the relationship between the entity match rate and the accuracy during training, as illustrated in the right panel of Figure 3. A strong positive correlation is evident: for both GRPO and E-GRPO, the curves of the entity match rate and accuracy rise in tandem. This confirms our core hypothesis that **the entity match rate serves as an effective proxy for final answer accuracy**. Crucially, the plot reveals the direct impact of our entity-aware reward: by explicitly incentivizing a higher entity match rate, E-GRPO (orange) consistently outperforms the GRPO baseline (green) on this metric. This advantage, in turn, directly translates into superior final answer accuracy (purple vs. blue), validating that **mastering the sub-goal of entity matching leads to better overall performance**.

A detailed case study in Appendix D provides a qualitative illustration of these dynamics, concretely demonstrating how E-GRPO's focus on entity matching leads to a more efficient and accurate reasoning path. Through the case study, we also analyze several failure cases of E-GRPO in Appendix E.

**Ablations of Entity Matching Weights.** We conduct an ablation study on the hyperparameter $\alpha$, which balances the outcome-based reward and the entity-matching bonus. As shown in Figure 4, setting $\alpha = 0.0$ reduces our method to the GRPO baseline. For all four benchmarks, performance consistently improves as $\alpha$ increases from 0.0, peaking at 0.3. This clearly demonstrates the benefit of incorporating the entity-aware reward. However, a further increase to $\alpha = 0.5$ leads to a performance drop on most benchmarks, suggesting that an excessively strong entity-matching bonus can distract the model from the primary goal of generating a correct final answer. This highlights the importance of balancing the two reward components, with a moderate $\alpha$ value yielding the optimal

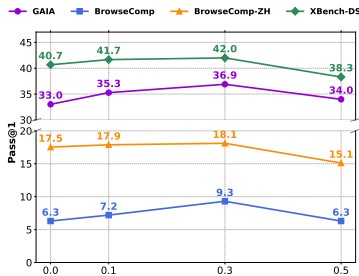

Figure 4: Comparison of different entity matching weights.

policy. More analysis of a decaying $\alpha$ value during training is available in Appendix F.

## 5 RELATED WORK

**Search Agents.** The capabilities of Large Language Models (LLMs) have fueled a surge in research on autonomous agents that can interact with external environments to solve complex tasks.

A foundational paradigm in this area is the ReAct framework (Yao et al., 2023), which interleaves reasoning (thought) and action steps. Building on this, a prominent line of research has focused on search agents designed to navigate the web (Song et al., 2025; Zheng et al., 2025; Li et al., 2025c; Zhang et al., 2025a; Sun et al., 2025). Advanced models like Gemini Deep Research (Google Team, 2025b), OpenAI Deep Research (OpenAI, 2025), Grok DeepSearch (xAI Team, 2025), along with smaller open-source models such as Asearcher (Gao et al., 2025), WebThinker (Li et al., 2025d), WebWalker (Wu et al., 2025b), WebDancer (Wu et al., 2025a), and WebSailor (Li et al., 2025b;a) have demonstrated increasing proficiency in retrieving and synthesizing information from noisy, real-world web sources. Our work directly contributes to improving the training methodology for this class of agents, addressing the challenge of learning robust policies in complex web environments.

**Synthetic Data Generation for Search Agents.** The now-dominant paradigm for training search agents relies heavily on high-quality synthetic data (Team, 2025b). A common thread in these generation methods is an entity-centric approach to complexity generation (Gao et al., 2025; Wu et al., 2025b;a; Li et al., 2025b;a; Tao et al., 2025; Wu et al., 2025c). During data synthesis, a rich set of ground-truth entities that form the factual backbone of the correct answer are systematically discarded. Prior work has exclusively used the final question-answer pairs from this process for post-training (Dong et al., 2025; Wu et al., 2025a; Li et al., 2025b). In contrast, our work is the first, to our knowledge, to recognize these discarded entities not as a byproduct, but as an untapped source of fine-grained, factual supervision. We pioneer the idea of repurposing this "waste" material to formulate an entity-aware reward function, thereby bridging the gap between the data generation process and the RL alignment phase in a novel and efficient manner.

**Reinforcement Learning for Search Agents.** Group Relative Policy Optimization (GRPO) and its variants (Shao et al., 2024; Yu et al., 2025; Xu et al., 2024; Zhao et al., 2025; Hu, 2025; Xue et al., 2025; Su et al., 2025) have become a dominant paradigm for aligning search agents. Notable advancements within this paradigm, such as ARPO (Dong et al., 2025), have adapted the framework with an entropy-based rollout mechanism for complex multi-turn web search settings. Despite these refinements, the entire family of GRPO-like methods is fundamentally constrained by its reliance on a sparse, outcome-based reward signal. While conventional solutions to such sparsity, like Process Reward Models (PRMs) (Fan et al., 2025; Anonymous, 2025; Zhang et al., 2025b) or tree-based sampling (Yang et al., 2025; Hou et al., 2025), exist in related domains, they are ill-suited for open-ended web search due to prohibitive annotation costs and computational intractability. Our work, E-GRPO, diverges from these approaches by proposing a reward signal that is both fine-grained and computationally efficient, requiring no additional annotation, model training, or complex sampling.

## 6 CONCLUSION

In conclusion, we propose Entity-aware Group Relative Policy Optimization (E-GRPO), a novel framework designed to enhance policy optimization for search agents. Our analysis reveals that the ground-truth entities discarded during synthetic data generation serve as a powerful proxy for factual correctness, offering a fine-grained reward signal that standard methods ignore. E-GRPO leverages this insight by formulating an entity-aware reward function, assigning partial credit to negative samples based on their entity match rate to encourage meaningful exploration. Across a wide array of QA and deep research benchmarks, E-GRPO consistently and significantly outperforms the GRPO baseline. Remarkably, it not only achieves superior accuracy but also learns more efficient policies with fewer tool calls, offering a more effective and sample-efficient solution for aligning search agents in complex, knowledge-intensive tasks.

## 7 ACKNOWLEDGMENT

This work was supported by Alibaba Group through Alibaba Innovative Research Program, the Core Facility Platform of Computer Science and Communication, SIST, and the HPC platform of ShanghaiTech University.

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

## A   FORMAT

Our ReAct framework follows Li et al. (2025b). A complete rollout follows the format below:

```
Format

<think> thinking process here </think>
<tool_call>
{"name": "tool name here", "arguments": {"parameter name here": parameter value here,
"another parameter name here": another parameter value here, ...}}
</tool_call>
<tool_response>
tool_response here
</tool_response>
(more thinking processes, tool calls and tool responses here)
<think> thinking process here </think>
<answer> answer here </answer>
```

Any response that does not strictly follow the format will be considered a case with format errors.

## B   DISCUSSION ABOUT SYNTHETIC-DATA ENTITIES

### B.1   ENTITY CONSTRUCTION

First, we explain the construction of the ground-truth entity sets for two different data synthesis methods used in our experiments.

**ASearcher.**   As illustrated in Figure 1 and Section 2.2, the question is iteratively constructed by selecting an entity and replacing it with descriptive facts or fuzzing it. Consequently, we use all selected and modified entities for a question as its ground-truth entity set.

**SailorFog-QA.**   As shown in Section 2.2, data generation begins by sampling an entity subgraph, followed by prompting an LLM to generate a question centered around these entity nodes. Therefore, the node set of the sampled subgraph is regarded as the ground-truth entity set.

**Entity Quality Control.**   Since the question generation process ensures question quality, e.g., injected facts strictly adhere to the selected entity, and generated questions are consistently centered around the sampled subgraph, the resulting entity sets are highly precise. Even if there are unexpectedly noisy entities, our reward mechanism is robust to them. Since any irrelevant entity is likely to be missed by all rollouts within a group, it does not change their relative performance and thus does not affect the normalized reward signal.

### B.2   ENTITY MATCHING

Then, we consider two questions related to the entity matching mechanism: (1) Why do we use the exact string match rather than using an LLM for matching? (2) Why do we only count the entities matched in thoughts, excluding those matched in observation?

**Rationale for Exact String Matching.**   Our decision to use exact string matching instead of an LLM-based judger is primarily motivated by the nature of our ground-truth entities, which are definite, short-formed strings with little ambiguity. This characteristic makes exact matching a natural and sufficient method, which in turn addresses two practical concerns: training efficiency and robustness against reward hacking.

First, employing an LLM to semantically parse and match entities within long reasoning traces would introduce significant computational latency, impeding the throughput of the RL training loop. In contrast, exact string matching is computationally trivial and adds negligible overhead.

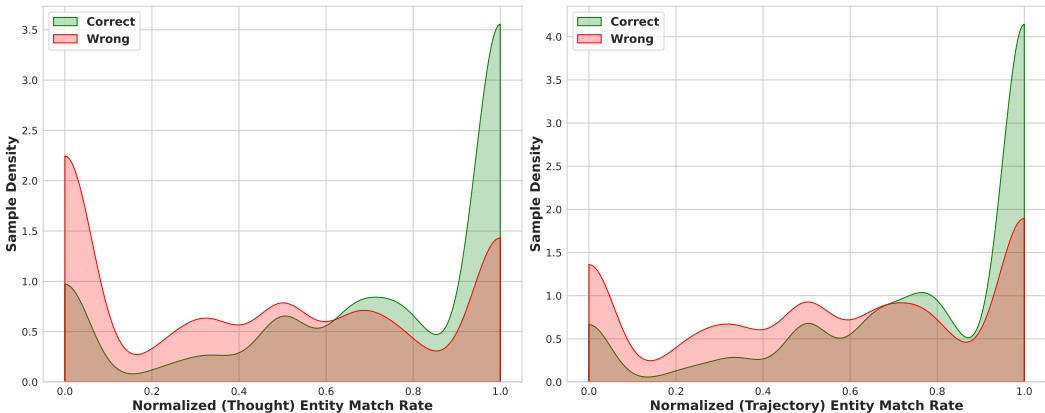

Figure 5: Comparison of Normalized entity match rate in thoughts and entire trajectories.

Second, while advanced LLMs can perform semantic matching, they are also more susceptible to exploitation by the policy model. In preliminary experiments, we observed a distinct reward-hacking behavior: the agent learned to extend its thoughts with verbose, superficially relevant phrases that, while not containing the correct entities, would mislead the LLM judge into erroneously assigning partial credit. Exact string matching, being less flexible, provides a more reliable reward signal, ensuring the agent is rewarded for factual correctness rather than plausible-sounding text.

**Rationale for Thought-Only Matching.** To justify why we match entities exclusively within the agent's thoughts (`<think> </think>` blocks), we analyze the difference between this approach and matching across the entire trajectory (including observations).

As shown in Figure 5, the two methods yield notably different distributions for incorrect trajectories. While thought-based matching (left) shows a clear separation with most failures having a low match rate, trajectory-based matching (right) produces significantly more "false positives": incorrect rollouts that still achieve a high entity match rate.

We do several case studies and find the cause of this discrepancy. Often, a key entity is present in the observation returned by a tool (e.g., a search snippet), but the agent fails to extract and incorporate this information into its reasoning process. Rewarding the agent based on the entire trajectory would grant unearned credit for merely encountering information, not for understanding and acting upon it. This creates a noisy reward signal that fails to penalize a true reasoning failure. Therefore, by confining entity matching to the agent's thoughts, we ensure the reward is directly coupled to the model's ability to identify and internalize key information, providing a cleaner and more targeted learning signal.

## C  HYPERPARAMETERS

**SFT.** We apply a training batch size of 32, a cosine decay learning rate of 5e-6 for about 4.8 epochs, with a linear warm up starting from 1e-10 for 0.2 epochs.

**RL.** We apply a context length of 32k, a tool call budget of 40, a training batch size of 64, a ppo batch size of 64 (exactly **on policy** optimization), a group size of 8, and a learning rate of 2e-6 for about 5 epochs. For E-GRPO, we set the entity matching weight $\alpha$ as 0.3 by default.

**Inference.** We apply a `temperature` of 0.6, a `top_p` of 0.95, and a tool call budget of 40 for inference.

## D CASE STUDY

To illustrate the qualitative differences in reasoning, we present a comparative case study using trajectories generated by Web-30B-GRPO and Web-30B-E-GRPO for the same query. The E-GRPO agent demonstrates a more direct and efficient reasoning path, requiring fewer tool calls to successfully identify all key entities and culminating in the correct answer. In stark contrast, the GRPO agent embarks on a longer, less fruitful trajectory. Despite using more tool calls, it fails to pinpoint the critical entity in the thoughts, the **First International Polar Year** (described as the "*international scientific initiative focused on the polar regions*"), which ultimately leads to an incorrect answer.

---

**Case Trajectory on SailorFog-QA: Solved by Web-30B-E-GRPO**

**Question:** A European naval officer, known for leading a significant polar expedition in the late 19th century aboard a vessel named after an Austrian admiral, later proposed a major international scientific initiative focused on the polar regions. For his contributions to exploration, he received one of the highest honors from a major British geographical society. Who is this explorer?
**Ground-truth Entities:** [Tegetthoff, International Polar Year, Royal Geographical Society's Founder's Medal]
**Answer:** Karl Weyprecht

**Round 1**

`<think>`
To identify the explorer in question, I need to connect several specific clues: a European naval officer who led a notable polar expedition in the late 19th century aboard a ship named after an Austrian admiral, later proposed a major international polar scientific initiative, and received a top honor from a major British geographical society. These are highly distinctive details, but the identity is not immediately clear. By structuring my search queries to include all these elements—expedition timing, vessel name origin, scientific initiative, and British recognition—I maximize the chance of surfacing sources that directly link these facts to a single individual. This comprehensive approach should yield either the explorer's name or authoritative references that will allow me to deduce the answer. I need to run these targeted searches now to synthesize the relevant information.
`</think>`
`<tool_call>`
{"name": "search", "arguments": {"query": ["European naval officer polar expedition late 19th century vessel named after Austrian admiral", "European naval officer proposed major international scientific initiative polar regions", "European naval officer received highest honors major British geographical society", "late 19th century vessel named after Austrian admiral polar expedition"]}}
`</tool_call>`

`<tool_response>`
A Google search for 'European naval officer polar expedition late 19th century vessel named after Austrian admiral' found 10 results:
## Web Results 1. [Austro-Hungarian North Pole expedition - Wikipedia]
(https://en.wikipedia.org/wiki/Austro-Hungarian_North_Pole_expedition) The main ship was the Tegetthoff , named for the Austrian Admiral Wilhelm von Tegetthoff, under whom Weyprecht had served. It was built for this expedition by Teklenborg & Beurmann in Bremerhaven. It was a three-masted barkentine of 220 tons, 38.34 m long, with a 100 horsepower (75 kW) steam engine.
2. [Wilhelm von Tegetthoff - Wikipedia]
(https://en.wikipedia.org/wiki/Wilhelm_von_Tegetthoff)
...
`</tool_response>`

**Round 2**

---

**<think>**
The Wikipedia article on the Austro-Hungarian North Pole expedition is likely to provide authoritative details about the expedition's leader, the ship's name, and its namesake. Since the question hinges on connecting the identity of the naval officer, the vessel named after an Austrian admiral, and subsequent achievements, reviewing this page should clarify who led the expedition and confirm the link between the ship and Admiral Wilhelm von **Tegetthoff**. Additionally, the article may mention the leader's later scientific initiatives and honors, which are crucial for fully answering the question. I need to visit this page to extract these specific details and establish the explorer's identity.
**</think>**
**<tool_call>**
{"name": "visit", "arguments": {"url": ["https://en.wikipedia.org/wiki/Austro-Hungarian_North_Pole_expedition"], "goal": "Identify the commander (led by) and the vessel's namesake name, Admiral Wilhelm von Tegetthoff"}}
**</tool_call>**

**<tool_response>**
The useful information in https://en.wikipedia.org/wiki/Austro-Hungarian_North_Pole_expedition for user goal Identify the commander (led by) and the vessel's namesake name, Admiral Wilhelm von Tegetthoff as follows:
Evidence in page: The Austro-Hungarian North Pole expedition was an Arctic expedition to find the North-East Passage that ran from 1872 to 1874 under the leadership of Julius von Payer and Karl Weyprecht. The expedition discovered and partially explored Franz Josef Land.
...
**</tool_response>**

### Round 3

**<think>**
To determine whether Karl Weyprecht received one of the highest honors from a major British geographical society, I need to directly associate his name with such recognition. The previous information confirmed his leadership of the expedition and the vessel's namesake, but the question also requires evidence of prestigious British honors. By searching for combinations of "Karl Weyprecht" with terms like "highest honors" and "British geographical society," I can efficiently uncover whether he was awarded something like the Founder's Medal or a similar distinction from the Royal Geographical Society. This targeted search should clarify if Weyprecht fits all aspects of the question. I need to run these specific queries now to confirm the final piece of information.
**</think>**
**<tool_call>**
{"name": "search", "arguments": {"query": ["Karl Weyprecht highest honors British geographical society", "Karl Weyprecht British geographical society", "Karl Weyprecht Founder's Medal", "Karl Weyprecht honors Royal Geographical Society"]}}
**</tool_call>**

**<tool_response>**
A Google search for 'Karl Weyprecht highest honors British geographical society' found 10 results:
## Web Results 1. [Karl Weyprecht - Wikipedia]
(https://en.wikipedia.org/wiki/Karl_Weyprecht) Missing: highest honors
...
**</tool_response>**

### Round 4

**<think>**
The search results strongly suggest that Karl Weyprecht is the central figure in question, with multiple sources mentioning his leadership of the 1872–74 polar expedition aboard

the **Tegetthoff** and his later scientific initiatives. However, to fully confirm his identity and gather all the necessary details—such as the ship's name, the Austrian admiral it was named after, his role in proposing the **International Polar Year**, and whether he received the **Royal Geographical Society's Founder's Medal**—I need a comprehensive, authoritative summary. Visiting the Wikipedia page dedicated to Karl Weyprecht should provide a consolidated account of his achievements, honors, and scientific contributions. I need to visit this page now to extract these specific details and definitively answer the question.
**</think>**
**<tool_call>**
{"name": "visit", "arguments": {"url": ["https://en.wikipedia.org/wiki/Karl_Weyprecht"], "goal": "Extract details: vessel name, the Austrian admiral, international scientific initiative, Founder's Medal of Royal Geographical Society"}}
**</tool_call>**

**<tool_response>**
The useful information in https://en.wikipedia.org/wiki/Karl_Weyprecht for user goal Extract details: vessel name, the Austrian admiral, international scientific initiative, Founder's Medal of Royal Geographical Society as follows:
Evidence in page: He co-led, with Julius von Payer, the 1872-1874 [[Austro-Hungarian North Pole Expedition]] which discovered the archipelago [[Franz Josef Land]] in the Arctic Ocean. The expedition's ship "Admiral Tegetthoff"...
**</tool_response>**

---

**Round 5**

**<think>**
To arrive at the answer "Karl Weyprecht," I pieced together several key clues from the question and verified them through targeted searches and authoritative sources. First, I confirmed that Weyprecht led a late 19th-century polar expedition aboard the ship "Admiral **Tegetthoff**," which was named after Austrian Admiral Wilhelm von **Tegetthoff**—this directly matched the ship and admiral mentioned in the question. Next, I established that Weyprecht was instrumental in proposing the **International Polar Year**, a major collaborative scientific initiative for the polar regions. Finally, I verified through multiple sources that he was awarded the 1875 **Royal Geographical Society's Founder's Medal**, one of the highest honors in British geography. The convergence of these facts—leadership of the Tegetthoff expedition, advocacy for the International Polar Year, and receipt of the RGS Founder's Medal—confirms that Karl Weyprecht is the explorer described.
**</think>**
**<answer>**
**Karl Weyprecht** exemplifies the characteristics in your query: as commander of a significant polar expedition (aboard the ship "Admiral **Tegetthoff**") after a prominent Austrian admiral, as a proposal target of a major international scientific initiative (the **International Polar Year**), and as a recipient of one of the most distinguished honors from a major British geographical society (the **Royal Geographical Society's Founder's Medal**).
**</answer>** ✓

---

**Case Trajectory on SailorFog-QA: Failed by Web-30B-GRPO**

**Question:** A European naval officer, known for leading a significant polar expedition in the late 19th century aboard a vessel named after an Austrian admiral, later proposed a major international scientific initiative focused on the polar regions. For his contributions to exploration, he received one of the highest honors from a major British geographical society. Who is this explorer?
**Ground-truth Entities:** [Tegetthoff, International Polar Year, Royal Geographical Society's Founder's Medal]
**Answer:** Karl Weyprecht

**Round 1**

**<think>**
The question involves a European naval officer who led a significant polar expedition in the late 19th century, had a vessel named after an Austrian admiral, later proposed a major international scientific initiative for the polar regions, and was honored by a prominent British geographical society. To pinpoint the individual, I need to connect the dots between the expedition, the vessel's name, the Austrian admiral reference, and the scientific initiative. Using search queries that combine these elements—such as the vessel's naming, the Austrian admiral, the polar expedition, and the proposal of an international scientific initiative—should surface historical records or articles that directly mention the explorer and clarify his contributions. By searching with these targeted terms, I aim to identify the officer and gather the necessary context for the rest of the question. I need to conduct this search now to find the explorer's identity.
**</think>**
**<tool_call>**
{"name": "search", "arguments": {"query": ["vessel named after Austrian admiral polar expedition late 19th century", "proposal major international scientific initiative polar regions", "later proposed major international scientific initiative polar regions"]}}
**</tool_call>**

**<tool_response>**
A Google search for 'vessel named after Austrian admiral polar expedition late 19th century' found 10 results:
## Web Results
1. [Austro-Hungarian North Pole expedition - Wikipedia]
(https://en.wikipedia.org/wiki/Austro-Hungarian_North_Pole_expedition) The main ship was the Tegetthoff , named for the Austrian Admiral Wilhelm von Tegetthoff, under whom Weyprecht had served. It was built for this expedition by Teklenborg & Beurmann in Bremerhaven.
...
**</tool_response>**

**Round 2**

**<think>**
The Wikipedia page for the Austro-Hungarian North Pole expedition is likely to contain authoritative details about the expedition's leadership, the role of the vessel named after an Austrian admiral, and key figures involved. Since the question requires identifying the naval officer who led this expedition, as well as his later contributions and honors, this page should provide not only his name but also contextual information about his career and achievements. By visiting this URL, I can directly extract the name of the expedition leader and see if there are references to his subsequent actions, such as proposing major scientific initiatives and receiving recognition from British geographical societies. I need to visit this page now to gather these crucial details.
**</think>**
**<tool_call>**
{"name": "visit", "arguments": {"url": ["https://en.wikipedia.org/wiki/Austro-Hungarian_North_Pole_expedition"], "goal": "Identify the leader of the expedition and aircraft details including his name, and whether he proposed a Polar Year or similar initiative later, and awards"}}
**</tool_call>**

**<tool_response>**
The useful information in https://en.wikipedia.org/wiki/Austro-Hungarian_North_Pole_expedition for user goal Identify the leader of the expedition and aircraft details including his name, and whether he proposed a Polar Year or similar initiative later, and awards as follows:

Evidence in page: The Austro-Hungarian North Pole expedition was an Arctic expedition to find the North-East Passage that ran from 1872 to 1874 under the leadership of Julius von Payer and Karl Weyprecht.

...
**</tool_response>**

---

**Round 3**

**<think>**
Although I have identified Julius von Payer as a leader of the Austro-Hungarian North Pole expedition aboard the **Tegetthoff** and found that the expedition contributed to the idea of international polar scientific collaboration, I have not yet pinpointed the exact honor from a British geographical society. This is a crucial detail needed to fully answer the question and distinguish his achievements from those of his contemporaries. By searching with focused queries such as 'Julius von Payer British Geographical Society honors', and 'Weyprecht British Geographical Society honors', I can directly target sources that clarify the connection between the expedition leaders and the specific honors received. I need to conduct this search now to fill these key information gaps.
**</think>**
**<tool_call>**
{"name": "search", "arguments": {"query": ["Julius von Payer British Geographical Society honors", "Weyprecht British Geographical Society honors"]}}
**</tool_call>**

**<tool_response>**
A Google search for 'Julius von Payer British Geographical Society honors' found 10 results:
## Web Results 1. [Julius von Payer - Wikipedia]
(https://en.wikipedia.org/wiki/Julius_von_Payer) He is chiefly known for the Austro-Hungarian North Pole expedition in 1872–74 and the discovery of Franz Josef Land ...
**</tool_response>**

**Round 4**

**<think>**
To fully confirm whether Julius von Payer received one of the highest honors from a major British geographical society, and to directly tie him to any significant international scientific initiatives for the polar regions, I need authoritative biographical details. The Wikipedia page dedicated to Julius von Payer is likely to provide a concise summary of his major achievements, awards, and any notable proposals or initiatives he was responsible for. By visiting this page, I can efficiently extract explicit references to British geographical society honors—such as the **Royal Geographical Society's Founder's Medal** or Patron's Medal—and clarify his role in shaping international polar scientific collaboration. I need to visit this page now to gather these definitive details.
**</think>**
**<tool_call>**
{"name": "visit", "arguments": {"url": ["https://en.wikipedia.org/wiki/Julius_von_Payer"], "goal": "Extract awards of Julius von Payer, particularly any from British Geographical Society, and clarify his role in shaping international polar scientific collaboration"}}
**</tool_call>**

**<tool_response>**
The useful information in https://en.wikipedia.org/wiki/Julius_von_Payer for user goal Extract awards of Julius von Payer, particularly any from British Geographical Society, and clarify his role in shaping international polar scientific collaboration as follows: ...
**</tool_response>**

**Round 5**

**<think>**
To determine which expedition leader—Julius von Payer or Karl Weyprecht—received the Royal Geographical Society's highest honors, I need to clarify whether Payer was actually awarded the gold medal or the Patron's Medal, and in which year. The summary on the main Julius von Payer page referenced the Patron's Medal in 1971, but individual award listings on the Gold Medal (RGS) Wikipedia page should provide a definitive, year-by-year roster of recipients, ensuring accuracy regarding both the medal and its recipients. By visiting this page, I can confirm the exact year and whether Julius von Payer is indeed listed among the honorees, which is crucial for answering the question about his recognition by the British geographical society. I need to check this official list now to confirm the details.
**</think>**
**<tool_call>**
{"name": "visit", "arguments": {"url": ["https://en.wikipedia.org/wiki/Gold_Medal_(RGS)"], "goal": "Check if Julius von Payer is listed as a recipient of the Royal Geographical Society's highest honor and year"}}
**</tool_call>**

**<tool_response>**
The useful information in https://en.wikipedia.org/wiki/Gold_Medal_(RGS) for user goal Check if Julius von Payer is listed as a recipient of the Royal Geographical Society's highest honor and year as follows:
Evidence in page: 1875 —[[Julius Payer]] —For explorations and discoveries in the Arctic regions ...
**</tool_response>**

### Round 6

**<think>**
To arrive at the answer, I first noticed that the question described a European naval officer who led a significant polar expedition in the late 19th century aboard a vessel named after an Austrian admiral, later proposed a major international scientific initiative, and received a top honor from a British geographical society. My initial searches confirmed that the Austro-Hungarian North Pole expedition (1872–1874), led by Julius von Payer and Carl Weyprecht, used the ship **Tegetthoff**, named after Admiral Wilhelm von **Tegetthoff**. Further investigation revealed that while both men were key figures, it was Julius von Payer who was awarded the Royal Geographical Society's Patron's Medal (the highest British geographical honor) for his achievements. Therefore, by piecing together the expedition details, the naming of the vessel, and the specific recognition from the RGS, I concluded that the explorer in question is Julius von Payer.
**</think>**
**<answer>**
**Julius von Payer** stands out as the notable European naval officer who fits all aspects of your query, having led a landmark polar expedition, played a key role in the foundation of international polar science initiatives, and been honored by the foremost British geographical society for his achievements.
**</answer>** ✗

## E    FAILURE ANALYSIS

Our extensive case study reveals that, in addition to the factual errors our paper targets at, agent failures frequently stem from several other distinct error modes, which we detail below.

- **Broken Reasoning Coherence**: The agent loses reasoning coherence by failing to act on its previously stated goal, particularly after a **Visit** tool call.

    Example:

    ```
    <tool_call>
    {"name": "visit", "arguments": {"url": "https://en.wikipedia.org/wiki/Teatro_Farnese",
    "goal": "Extract notable architectural features, innovative aspects, and check
    perspective scenery introduction"}}
    </tool_call>
    <tool_response>
    The useful information...a theatre in which the audience views the action through a
    single frame...
    </tool_response>
    <think> To determine whether the Teatro alla Scala is relevant to the question or if
    it is merely a famous modern opera house... </think>
    ```

    In this example, the agent exhibits a loss of reasoning coherence. While it correctly visits the page and extracts a key fact ("a single frame"), it subsequently ignores its own goal for that visit ("notable architectural features") and instead begins reasoning about an unrelated subject.

- **Distracted Querying**: The agent loses focus on the core objective and instead pursues secondary information triggered by minor details, deviating from the optimal solution path, finally exceeding the context length.

    Example:

    ```
    Question: ...What year saw the first performance of this multi-screen instrument
    by the aforementioned performer?
    <tool_call>
    {"name": "search", "arguments": {"query": ["multi-screen instrument designed
    early 1970s performer Charlotte Moorman"]}}
    </tool_call>
    ...
    <tool_call>
    {"name": "search", "arguments": {"query": ["MoMA instrument multi screen"]}}
    </tool_call>
    ```

    In this example, the agent fails to maintain its core objective. It correctly formulates an early query that includes the temporal aspect ("early 1970s"), but then gets distracted by other entities ("MoMA"). The agent forgets that its primary task was to find the year, as shown by the absence of any time-related keywords in its later search query.

- **Information Overload from Concurrent Queries**: By executing multiple search queries within a single step, the agent generates a large volume of unstructured text. This information overload overwhelms its processing capacity, causing it to fail to identify or retain the most critical facts necessary for the subsequent reasoning step.

    Example:

```
<tool_call>
{"name": "search", "arguments": {"query": ["appointed board central banking sys-
tem 2016", "appointed serve on board central banking system appointed 2016", "ap-
pointed board central banking system later became Chair 2016"]}}
</tool_call>
```

In this example, the agent executes three highly similar queries simultaneously. While only the third query is well-targeted, containing the critical term "Chair", the valuable information it returns is buried within the larger volume of text from all three searches. Overwhelmed by the dense context, the agent fails to isolate the key signal from the noise, which misleads it down a flawed reasoning path.

Our case study indicates E-GRPO mitigates **broken reasoning coherence**. By incentivizing the agent to carry forward critical entities into its thought process, the entity-aware reward implicitly enforces topical continuity and prevents the reasoning from drifting into irrelevance.

The other identified failure modes, however, suggest promising avenues for future work:

- For **distracted querying**, a potential solution is to periodically re-inject the original question into the agent's context, orienting the agent's focus towards the primary objective.
- We attribute **information overload** primarily to the inherent limitations of the base model in processing long, unstructured contexts. Addressing this could involve either leveraging more advanced models with superior long-context capabilities or designing a penalty mechanism for imprecise or redundant queries.

## F    DECAYING ENTITY MATCHING WEIGHTS

| alpha | GAIA | BrowseComp | BrowseComp-ZH | xbench-DS |
|-------|------|------------|---------------|-----------|
| 0.3   | 48.5 | 12.9       | 26.4          | 46.7      |
| decay | 48.2 | 12.8       | 26.2          | 47.3      |

Table 3: The Pass@1 performance of different entity matching weights.

We also train the 30B model with an alpha that linearly decays from 0.3 to 0.0 over the first 60 steps (80 steps in total). The results are presented in Table 3.

Comparing with a fixed 0.3, the results show no clear or consistent advantage for the decaying alpha strategy. This suggests our paper's choice of a simple, fixed alpha as a practical and effective setting. However, we believe a dynamic alpha holds potential. It's possible that applying the decaying schedule over longer training horizons could be more impactful, which is a promising direction for future research.

## G    EXTENDED TRAINING DYNAMICS

We extend the training of 30B models from 80 steps to 120 steps with the Web environment. As shown in Figure 6, the results align with our analysis in Section 4.3. While both methods' performance gradually converges, E-GRPO maintains a consistent lead over GRPO. Crucially, E-GRPO also continues to use fewer steps on average. This demonstrates E-GRPO's dual advantages in both effectiveness and efficiency.

## H    ROBUST MATCHING ANALYSIS

While exact string matching is computationally efficient and effectively mitigates reward hacking (see Appendix B.2), its rigidity presents practical challenges when faced with natural language variations such as alternative spellings, synonyms, or abbreviations. To address this limitation, we

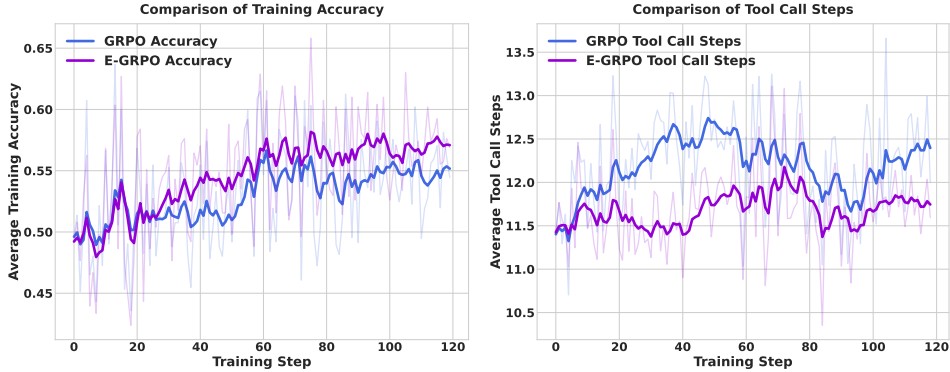

Figure 6: Extended training dynamics of 30B models with the Web environment.

introduce a "Robust Matching" strategy. This approach involves first prompting an advanced LLM to generate a set of 5-10 plausible variations for each ground-truth entity, as detailed in the prompt below. Subsequently, we perform exact string matching against this expanded set of candidates. An entity is matched if any of its generated variations are found in the agent's thought process.

---

**Prompt**

I need to match an entity in a text, but I need to account for various paraphrases, abbreviations, and potential spelling variations. For each of the entities listed below, please provide 5-10 alternative phrasings or variations that I can use for a more robust matching system. When generating the alternatives, please consider:
1. Synonyms and Paraphrasing: Using different words or sentence structures to convey the same meaning.
2. Abbreviations: Common initials, acronyms, or shortened forms.
3. Spelling Variations: Including regional differences (e.g., US vs. UK English) or common typos.
4. Formality: Both formal and informal ways of referring to the entity.
Note that you do not have to include words like "the" or "a" in your responses in order for robust matching.
I will give you the entity, and you should directly return a list of alternative phrasings in a valid JSON list format like ["phrasing 1", "phrasing 2", ...], do not include any other text in your response.
Entity: {**entity**}

---

With this "Robust Matching" strategy, we conduct the analysis of entity matching and correctness. Figure 7 visualizes the efficacy of this strategy. Compared with exact string matching, the "Robust Matching" strategy demonstrates a much stronger correlation with factual correctness. Specifically, for correct samples, the density peak at a match rate of 1.0 is significantly higher. Conversely, for incorrect samples, the peak at 0.0 is reduced, and the distribution becomes more spread out, better capturing "near-misses".

This analysis confirms that "Robust Matching" serves as a more accurate proxy for factual correctness. Motivated by this finding, we conducted an additional RL experiment using the "Robust Matching" strategy, keeping all other settings identical to Web-30B-E-GRPO. The results are presented in Table 4.

| Strategy | GAIA | BrowseComp | BrowseComp-ZH | xbench-DS |
|---|---|---|---|---|
| Exact String Matching | 48.5 | 12.9 | 26.4 | 46.7 |
| Robust Matching | 49.2 | 12.9 | 26.8 | 47.0 |

Table 4: The Pass@1 performance of different entity matching strategies.

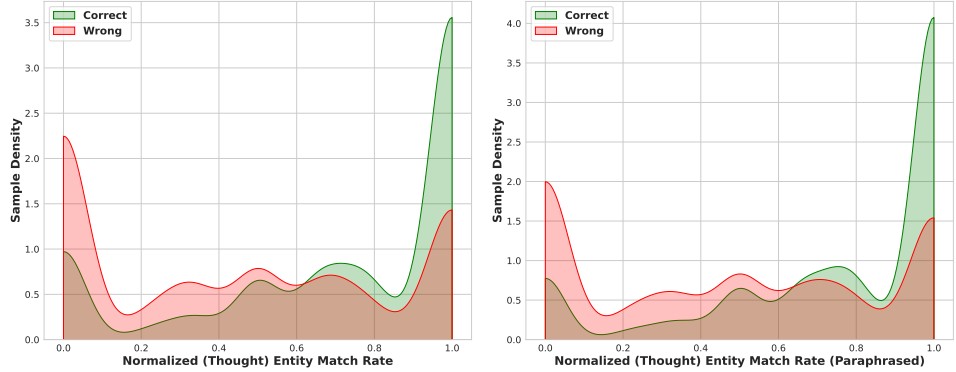

Figure 7: The entity match rate of exact string matching and robust matching.

The results are promising. The agent trained with "Robust Matching" achieves slight performance gains on three of the four deep research benchmarks, further validating the benefits of it. We consider the continued refinement of this strategy a promising direction for future work.

## I EXPLANATION OF THE ANALYSIS FIGURE

Here we clarify the upper-right subfigure of Figure 1. For a question $q$, we sample 8 responses $r_1, r_2, ..., r_8$ for it. Each response matches $m_i$ entities. We calculate the average matched entities of correct responses and that of incorrect ones. For example, if only $r_1, r_2, r_3$ are correct, then **correct rate** is $(m_1 + m_2 + m_3)/3$ and **wrong rate** is $(m_4 + m_5 + m_6 + m_7 + m_8)/5$. We compare these two rates for each question. The analysis shows that for most of the questions, correct rate is larger than wrong rate, i.e., **correct responses of each question often match more entities than wrong ones**. This establishes a strong correlation between the entity match rate and the correctness of the final answer.

## J USE OF LARGE LANGUAGE MODELS (LLMS)

In this paper, we leverage Gemini-2.5-Pro (Google Team, 2025a) to polish paper writing grounded on our human-written content. The usage is limited to language polishing. We are responsible for any contributions generated by LLMs.

