# OpenReview forum: "Repurposing Synthetic Data for Fine-grained Search Agent Supervision"
_ICLR.cc/2026/Conference — ICLR 2026 Poster_

### Official Review · Reviewer_LHoD · 2025-10-15

**Soundness:** 2
**Presentation:** 3
**Contribution:** 2
**Rating:** 4
**Confidence:** 4

**Summary:**

This paper introduces a reinforcement learning framework Entity-aware Group Relative Policy Optimization (E-GRPO), to enhance the training of LLM-based search agents. Existing GRPO methods usually rely on sparse, outcome-based rewards, treating all incorrect outputs equally and overlooking partially correct cases. In contrast, E-GRPO uses entity-centric information from synthetic training data by assigning partial rewards based on the entity match rate, i.e., the proportion of ground-truth entities correctly identified during reasoning. Experiments also show that entity-aware rewards provide a computationally efficient and semantically rich supervision signal for training search agents.

**Strengths:**

1.	Comprehensive experimental results are reported, including diverse LLM-based search agents and datasets.

2.	The experimental settings and evaluation details are clearly provided to facilitate reproducing.

3.	The paper is generally clearly-written and easy to follow.

**Weaknesses:**

1.	The empirical finding of “strong positive correlation between the number of ground-truth entities identified during the agent’s reasoning process and final answer accuracy” is relatively trivial and expectable. This simply follows the fact that correct reasoning paths naturally tend to include more correct entities.

2.	While the idea of entity-aware rewards is useful, the contribution seems an incremental improvement on GRPO rather than a fundamental paradigm shift. The core update is scaling reward by entity match rate, which is conceptually straightforward.

3.	In the introduction, it argues that PRM methods cannot handle open-ended, dynamic nature due to the need for step-wise annotation. However, the E-GRPO also needs the ground-truth entities for the complete reasoning path to compute the partial reward. This seems to suggest that E-GPRO also cannot handle the tasks that PRM cannot. The utility needs justification.

4.	The entity annotations in synthetic datasets are assumed correct and reliable. If entity extraction or labeling is noisy, the reward signal may mislead training. The paper could better address this potential sensitivity to data quality and generalization to non-synthetic data.

5.	The paper does not sufficiently address how well E-GRPO generalizes to real-world search questions that differ substantially from the synthetic data distribution.


Minor:

1.	In the abstract, “We address this by leveraging the very entities discarded during training.”, is the word “very” a typo?

2.	The related work section should be a part of the main content to make it self-contained, instead of appearing only in the appendix.

**Questions:**

1.	The notion of “ground-truth entities” in reasoning requires clarification. Are there cases where multiple reasoning paths are valid, and if so, how does the method define or handle ground-truth entities in those scenarios?

2.	How well does E-GRPO agent generalize to other **question** domains or verticals that differ from the synthetic training data?

---

> ### Author Response · Authors · 2025-11-22
> **Response to Reviewer LHoD**
>
> We sincerely appreciate the effort and time you have dedicated to evaluating our work.
>
> > Weakness 1 & Question 1: Trivial finding of the fact that correct reasoning paths naturally tend to include more correct entities. Questions about handling cases where multiple reasoning paths are valid.
>
> 1. The empirical analysis is a non-trivial prerequisite:
>
>     Thanks for your insightful questions and your question 1 highlights the motivation for our empirical analysis. You are right that it seems intuitive for correct reasoning to involve correct entities. However, **this is a crucial assumption that requires validation, especially given your second point about the existence of multiple valid reasoning paths**. If many diverse, successful paths existed that did not rely on a common set of entities, we would not observe a strong positive correlation between entity match rate and final accuracy. Therefore, our empirical analysis (Figure 1) is a non-trivial and necessary analysis. The strong correlation provides powerful evidence that, despite theoretical path diversity, **successful reasoning in practice converges on a core set of factual nodes (entities)**.
>
> 2. Valid reasoning paths should cover the critical entities:
>
>     First, from the empirical analysis mentioned above, different reasoning paths are highly likely to share the same set of entities. Secondly, considering the data synthesis process, **question and answer pairs are systematically generated from these entities**. To solve the problem, the agent must effectively recover these critical points.
>
> Finally, in the rare case that a correct path finds an alternative solution without mentioning these entities (e.g., first guessing then verifying), our method does not penalize it; it still receives the full success reward of 1 (same as GRPO). Since these cases are empirically infrequent, our method remains effective without being overly restrictive.
>
> > Weakness 2: The core update is scaling reward by entity match rate.
>
> We agree that our core contribution lies in a novel reward design technique. To more accurately reflect this, we will revise the descriptions of our method accordingly in the final version.
>
> > Weakness 3: What is the advantage of our method compared with PRM?
>
> The core differences of these two methods are cost and reliability:
>
> 1. Trained PRMs are Prohibitively Expensive and Unreliable.
>
>     Cost: Standard PRMs require **extensive, step-by-step annotations** to train a model that grades each reasoning step. For the long, complex reasoning paths in web search, creating such a dataset is computationally and financially infeasible.
>
>     Reliability: PRMs give **evaluative grading of each step**, which is inherently unreliable in a noisy and complex environment, and may not be robust to reward hacking.
>
> 2. E-GRPO is Zero-Cost and More Robust.
>
>     Zero-Cost: Our entity-aware reward requires **no extra annotation**. It elegantly repurposes byproducts of the synthetic data generation pipelines that now dominate agent training. This makes it a zero-cost, highly scalable solution.
>
>     Reliability: Instead of grading ambiguous individual steps, our method provides a **factual score for the entire reasoning path**. This holistic approach is more reliable, less susceptible to reward hacking, and better reflects the overall quality of the agent's reasoning process.
>
> In essence, while one could view our method as a form of heuristic or rule-based PRM, it fundamentally differs from the expensive, trained PRMs.
>
> > Weakness 4: The paper could better address this potential sensitivity to data quality and generalization to non-synthetic data.
>
> We agree that our training relies on synthetic data, which is a deliberate choice aligning with the current SOTA paradigm for agent training. The prohibitive cost of human annotation for complex reasoning makes synthetic data a necessity, as evidenced by leading works like Kimi K2 and Tongyi-Deepresearch. The key contribution of our methods is not whether to use synthetic data for training, but how to leverage it most effectively.
>
> Therefore, we do not mean to design a universally applicable reward function. The goal of E-GRPO is to produce a better and generalizable model with now-dominate synthetic data. As shown in our experiments (Table 2), this training method leads to a final model with superior generalization and performance in real-world, non-synthetic scenarios (GAIA, xbench-DeepSearch, BrowseComp...).

---

> > ### Author Response · Authors · 2025-11-22
> >
> > > Weakness 5 & Question 2: How does E-GRPO generalize to real-world questions that differ from the synthetic training data?
> >
> > Our data synthesis process is anchored in **entities from Wikipedia** (as mentioned in WebSailor and ASearcher). Our evaluation setup deliberately includes two types of benchmarks:
> >
> > 1. QA Benchmarks (In-Domain Generalization): Multi-hop/Single-hop QA can be considered "in-domain" as they are largely solvable using Wikipedia knowledge, which is similar to our data synthesis environment. The strong performance here validates our method's effectiveness on tasks aligned with the training data distribution.
> >
> > 2. **Real-World Search Benchmarks (Out-of-Distribution (OOD) Generalization)**: The most critical test of generalization comes from our deep research benchmarks (GAIA, BrowseComp(-ZH), xbench-DeepSearch). These benchmarks consist of complex, real-world questions created and annotated by humans, requiring open-web search and reasoning far beyond the scope of Wikipedia. They are fundamentally out-of-distribution (OOD) relative to our Wiki-based training data.
> >
> > The superior performance of our E-GRPO-trained agent on these challenging OOD benchmarks is the strongest evidence of its generalization capabilities. It demonstrates that the enhanced reasoning skills learned via E-GRPO successfully transfer to novel, real-world problems that differ significantly from the synthetic training data.
> >
> > We hope this explanation helps you better understand our data synthesis, training, and evaluation setting. We have improved the description in **Section 4.1** accordingly to make it clearer for your reference.
> >
> > > Minor Weakness 1: The use of 'very' in abstract.
> >
> > Thank you for the suggestion. We used 'very' in its adjectival sense, meaning 'exact' or 'particular', e.g., *You're the **very** reviewer we needed to help us improve our paper.*
> >
> > > Minor Weakness 2: The related work should appear in the main content.
> >
> > Thank you for the suggestion. We have moved this section into the main content in the rebuttal version.

---

> ### Author Response · Authors · 2025-11-25
> **Request for Feedback**
>
> We have tried our best to address the issues raised in your review. Could you please let us know if there are any remaining concerns?

---

### Official Review · Reviewer_4vB2 · 2025-10-25

**Soundness:** 2
**Presentation:** 2
**Contribution:** 2
**Rating:** 4
**Confidence:** 4

**Summary:**

This paper introduces E-GRPO that improves the performance of LLM-based search agents. E-GRPO extends the vanilla GRPO with a denser, entity-aware reward function. It reuses the ground-truth entities from the synthetic training data to assign partial credit to incorrect answers based on how many key entities they successfully identified. Experiments show this approach allows the model to learn more effectively, consistently outperforming the GRPO baseline in accuracy and learning more efficient policies that use fewer tool calls.

**Strengths:**

- The reward formulation originated from a good observation in the data and its design is clear and reasonable.
- The empirical performance appears to be strong and better than vanilla SFT & GRPO.

**Weaknesses:**

- The research contribution appears to be incremental. While E-GRPO seems to outperform GRPO, I am not fully convinced E-GRPO is a fundamentally “novel framework” compared to the original GRPO given the fact that it only customizes the reward function. Partial entity matching in RL is also an established method, especially in NL2SQL field (e.g. Reasoning-SQL).
- It is nuanced whether the performance comparison with other baseline models in the main results is a fair comparison using same training data.

**Questions:**

- It appears that the training datasets for E-GRPO and other baseline models are different because the performance scores are directly cited from those external publications. Can authors clarify how much performance improvement comes from discrepancy in training data?
- While uniform credit assignment is a common practice in many GRPO-like RL training algorithms, is it technically sound and mathematically stable in E-GRPO when the partial entity matching reward is also uniformly assigned to the tokens, regardless of whether that specific token's "turn" was the one that found an entity or the one that made a mistake?
- What is the accuracy trend of GRPO and E-GRPO after 80 steps in fig. 3? Is it possible that the training happened to stop at the point where the difference is large, given the large fluctuation in both training curves?

---

> ### Author Response · Authors · 2025-11-22
> **Response to Reviewer 4vB2**
>
> We sincerely appreciate the effort and time you have dedicated to evaluating our work.
>
> > Weakness 1: The work only customizes the reward function. Partial entity matching is established in NL2SQL.
>
> We acknowledge that our main contribution is in a reward design technique, but we want to argue that we are not "inventing" partial-match rewards, but are the first to identify and propose an elegant and efficient solution to a key bottleneck in training modern search agents with GRPO. Specifically, we repurpose the byproducts of the synthetic data generation process (i.e., entities) into a zero-cost, dense reward signal, which better bridges the data synthesis pipeline with the RL training loop.
>
> With regard to the NL2SQL field, the partial matching reward, such as in Reasoning-SQL, are typically based on the **structure of the final, structured output (the SQL query)**. They measure the syntactic or component-wise correctness of the outcome. They do **not explicitly reward the intermediate reasoning process**. Our work, in contrast, rewards entities matched within the **unstructured, free-form reasoning path (thoughts)**. Our goal is to explicitly incentivize **correct factual reasoning** and exploration in a highly dynamic and noisy web environment. This is different from validating a structured output.
>
> > Weakness 2 & Question 1: Request "fair" comparison with other baseline models using same training data. No discussion about the improvement from discrepancy in training data.
>
> First, we want to claim that our experimental design deliberately incorporates two distinct types of comparisons to serve two different purposes:
>
> 1. **A Fair Comparison at the Algorithmic Level**: All models trained with E-GRPO are trained with standard GRPO on the same training data to strictly and fairly validate the algorithmic advantage of E-GRPO over GRPO.
>
> 2. **A System-level Comparison for SOTA Benchmarking**: We compare our methods with all other work in order to position our training system within the current research landscape. We must emphasize that all these methods are primarily based on the ReAct paradigm, with only a few differences in the format and prompt. **The core contribution of these methods (such as ASearcher, WebDancer, WebSailor) is intrinsically tied to their unique training data**. Their "methods" are inseparable from their training data. Therefore, we acknowledge that calling them "baselines" is misleading and will revise this to "reference agents" in the paper.
>
> In summary, our experimental setup is deliberately designed to first prove our algorithm's value in a controlled setting, then place it in the broader SOTA landscape. Additionally, this two-tiered comparison also resolves your question 1: **by comparing the performance of our GRPO baselines and the results of other reference agents, one can infer the performance improvement from discrepancy in training data.**
>
>
> > Question 2: Whether uniformly assigned entity matching reward is technically sound and mathematically stable. Why not apply turn/token-level credit to entities?
>
> Our trajectory-level reward is both technically sound and more appropriate for this task.
>
> 1. On Stability: Our design is stable for two reasons. First, E-GRPO only modifies the reward, preserving the advantage normalization, so we do not change the stability of GRPO. Second, the entity bonus in [0, $\alpha$] often reduces reward variance compared to GRPO's sparse binary reward, which can actually improve learning stability.
>
> 2. On Credit Assignment: We avoid turn-level rewards because they can be misleading. A crucial reasoning step (e.g., planning the next move) might not mention any entities and would be unfairly penalized. A trajectory-level reward holistically assesses whether key facts were found, granting the agent more flexibility, rather than micromanaging each step.
>
> > Question 3: The accuracy trend of GRPO and E-GRPO after 80 steps is unknown.
>
> We have extended the training past 80 steps to 120 steps, with the full learning curves now available in **Appendix G** of our modified paper. The results show that while both methods' performance gradually converges, E-GRPO maintains a consistent lead over GRPO. Crucially, E-GRPO also continues to use fewer steps on average. This demonstrates E-GRPO's dual advantages in both effectiveness and efficiency.

---

> ### Author Response · Authors · 2025-11-25
> **Request for Feedback**
>
> We have tried our best to address the issues raised in your review. Could you please let us know if there are any remaining concerns?

---

### Official Review · Reviewer_fNgM · 2025-10-31

**Soundness:** 3
**Presentation:** 3
**Contribution:** 3
**Rating:** 6
**Confidence:** 5

**Summary:**

This paper addresses the reward sparsity problem in training LLM-based search agents with reinforcement learning. Current methods totally assign a zero reward even if the trajectory contains partially correct information. The authors use an entity-centric method to synthesize a dataset and save the entities as ground-truth information. Then they add an extra entity-matching reward to improve performance.

Contributions:
* Identifying the "near-miss" problem: The paper shows that standard GRPO fails to distinguish informative partially correct samples from complete failures in search-agent training.
* Entity-aware reward function: A novel reward formulation that assigns partial credit based on entity match rate

**Strengths:**

* **Well-motivated problem**. Although partial correctness has been studied in RL, introducing it in training search agents is a good problem to facilitate this community.
* **Technical soundness**. The solution using the entity-matching score is reasonable, and the analysis of the relationship between accuracy and the matching score supports the insights of the proposed method.

**Weaknesses:**

1. **Handling of incorrect reasoning with correct entities**. The entity-matching reward may inadvertently credit erroneous reasoning paths that happen to mention correct entities without proper understanding. For instance, a model might generate factually incorrect statements while coincidentally including the right entity names. The paper does not address how to distinguish between genuine entity identification and spurious mentions.
2. **Overclaim**. While the entity-aware reward is effective, the core contribution is to augment GRPO with an additional reward term. The branding as "E-GRPO" may overstate the methodological advance, as the fundamental GRPO framework remains unchanged. A more modest framing as a reward technique might be more appropriate.
3. **Insufficient baseline comparisons in controlled settings**. The Local environment experiments (Table 1) would benefit from comparisons with more approaches that are feasible in controlled settings.
4. **Unexplained performance degradation**. The results show E-GRPO slightly underperforms on PopQA. The paper does not analyze or explain this anomaly.
5. **Absence of systematic error taxonomy and failure analysis**. Given the lengthy reasoning traces, the paper would benefit from categorizing failure modes (e.g., entity identification errors, reasoning chain breaks, etc.). Besides, the paper does not provide any concrete failure cases of the E-GRPO. While the case study in Appendix E shows a successful E-GRPO trajectory compared to a failed GRPO one, there is no analysis of cases in which E-GRPO fails despite its entity-aware reward.
6. **Entity matching robustness concerns**. While the authors justify exact string matching, they do not address practical challenges such as spelling variations, abbreviations, etc. These issues could significantly degrade the quality of the reward signal in real-world applications.

**Questions:**

1. **Figure 1 (upper-right) requires clarification**. I cannot fully understand the upper-right panel of Figure 1, especially the meaning of the x-axis labels. I would like to request more details and straightforward explanations of it.
2. **Figure 1 (lower-right) distribution analysis**. In the lower-right panel, the distribution shows cases where the entity match rates are not 1 but still lead to correct answers. Are these instances of hallucination where the model just happened to get it right?

---

> ### Author Response · Authors · 2025-11-22
> **Response to Reviewer fNgM**
>
> We sincerely appreciate the effort and time you have dedicated to evaluating our work.
> > Weakness 1: Not handling incorrect reasoning with correct entities.
>
> We acknowledge the intrinsic trade-off in designing agent reward systems: while a perfect path involves both correct entities and reasoning, quantifying the correctness of free-form reasoning is difficult. In contrast, verifying entities is much easier.
>
> Therefore, E-GRPO uses entity identification as a scalable proxy for correct reasoning. The effectiveness of this proxy is empirically validated by our own analysis (Figure 1), which demonstrates a strong positive correlation between entity match rate and final answer accuracy.
>
> This strong correlation suggests that, in practice, **cases of erroneous reasoning paths that happen to contain correct entities are infrequent**. Consequently, rewarding for correct entities may also serve as an efficient heuristic for encouraging correct reasoning.
>
> > Weakness 2: Overclaim.
>
> We agree that our core contribution lies in a novel reward design technique. To more accurately reflect this, we will revise the descriptions of our method accordingly in the final version.
>
> > Weakness 3: Insufficient baseline comparisons in our 2024 wiki environment.
>
> Our experiment in the Local environment was primarily designed as a controlled study (in-domain and stable environment) to validate the algorithmic advantage of E-GRPO over GRPO.
>
> Most of the other agents are typically trained with the open-domain web environment or their own environment instead of our 2024 wiki local environment, and targeted at real-world web searching ability. Evaluating them in a local environment could lead to an unfair comparison.
>
> > Weakness 4: Unexplained performance degradation in PopQA.
>
> Thank you for this insightful question. We attribute the performance on PopQA to three factors:
>
> 1. The marginal difference between E-GRPO (50.2) and GRPO (50.4) falls within the range of normal statistical variance for this benchmark (1000 randomly sampled testcases).
> 2. E-GRPO is designed for complex reasoning tasks. Its advantage is naturally minimal on simple, single-hop QA like PopQA, which lacks intermediate reasoning steps.
> 3. Unlike the diverse, real-world queries in NQ and TQ where our method excels, PopQA uses rigid, unambiguous templates. This removes the need for the nuanced reasoning and disambiguation that may have been strengthened by our entity-aware reward.
>
> > Weakness 5: Absence of systematic error taxonomy and failure analysis.
>
> Thanks for your insightful point. We conducted an extended case study and identified three typical failure modes.
> 1. **Broken Reasoning Coherence**:  The agent loses reasoning coherence by failing to act on its previously stated goal, particularly after a **Visit** tool call.
> 2. **Distracted Querying**: The agent loses focus on the core objective and instead pursues information triggered by minor details, deviating from the optimal solution path, finally exceeding the context length.
> 3. **Information Overload from Concurrent Queries**: By executing multiple search queries within a single step, the agent generates a large volume of unstructured text. This information overload overwhelms its processing capacity, causing it to fail to identify the most critical facts necessary for the subsequent reasoning steps.
>
>
> We also provide three short examples for each in **Appendix E** of the modified paper for your reference, with the latter two representing the failure cases of both GRPO and E-GRPO.
>
> > Weakness 6: Entity matching robustness concerns.
>
> We agree that robustness to entity variations is crucial. To investigate this, we propose a "Robust Matching" strategy, where we expanded entities with multiple (5 to 10) LLM-generated variations. An entity is matched if any of its variations appear in the reasoning path.
>
> We analyze the entity matching of "Robust Matching", which demonstrates a stronger correlation with factual correctness compared with exact string matching. Motivated by its potential, we run RL experiments using the "Robust Matching" strategy and achieve slight performance gains on several deep research benchmarks.
>
> |Strategy|GAIA|BrowseComp|BrowseComp-ZH|xbench-DS|
> |-----|----|----------|-------------|---------|
> |Exact String Matching|48.5|12.9|26.4|46.7|
> |Robust Matching|49.2|12.9|26.8|47.0|
>
> We consider the continued refinement of this strategy a promising direction for future work. The detailed design, analysis (with figures), and experimental results are also presented in **Appendix H** for your reference.

---

> > ### Author Response · Authors · 2025-11-22
> >
> > > Question 1: Figure 1 (upper-right) requires clarification.
> >
> > We apologize for the ambiguity and will clarify this point in detail. For a question $q$, we sample 8 responses $r_1, r_2, ..., r_8$ for it. Each response matches $m_i$ entities. We calculate the average matched entities of correct responses and that of incorrect ones. For example, if only $r_1, r_2, r_3$ are correct, then **correct rate** is $(m_1+m_2+m_3)/3$ and **wrong rate** is $(m_4+m_5+m_6+m_7+m_8)/5$. We compare these two rates for each question. The analysis shows that for most of the questions, correct rate > wrong rate, i.e., **correct responses of each question often match more entities than wrong ones**. This establishes a strong correlation between the entity match rate and the correctness of the final answer.
> >
> > > Question 2: Figure 1 (lower-right) distribution analysis. What is the cause of correct cases where the entity match rates are not 1?
> >
> > Our analysis of these cases reveals that correct outcomes with an entity match rate below 1.0 primarily stem from two distinct phenomena:
> > 1. As you mentioned in weakness 6, these cases may contain variations of our entities, which are not matched due to spelling variations, abbreviations, etc.
> > 2. The agent may occasionally generate the correct final answer without relying on retrieved evidence. This can occur either through "correct hallucination" or simply by chance.

---

> ### Author Response · Authors · 2025-11-25
> **Request for Feedback**
>
> We have tried our best to address the issues raised in your review. Could you please let us know if there are any remaining concerns?

---

> > ### Comment · Reviewer_fNgM · 2025-11-27
> >
> > Thanks for your rebuttal. Most of my concerns have been addressed. I will keep my positive score. I kindly ask the authors to improve the presentation of Figure 1 in the final version to enhance its clarity.

---

> > > ### Author Response · Authors · 2025-11-30
> > >
> > > Thank you for your valuable feedback and for confirming that our rebuttal has addressed your concerns. We are very grateful for your positive assessment and support for our work. We have revised the rebuttal version to enhance the clarity of Figure 1 as you suggested.

---

### Official Review · Reviewer_oC3b · 2025-11-01

**Soundness:** 4
**Presentation:** 3
**Contribution:** 2
**Rating:** 6
**Confidence:** 3

**Summary:**

The paper proposes an improvement to search agent RL algorithms such as Group Relative Policy Optimization (GRPO), which treat so-called near-miss rollouts as bad as complete miss rollouts. The authors propose entity-awareness for the reward function, which assigns partial credit to rollouts, leading to a denser reward function.

The approach is evaluated for QA datasets such as TriviaQA or PopQA, and Deep Research datasets such as GAIA or BrowseComp. A local (Wikipedia) and web setting (Google Search, Jina) are evaluated. Baselines are GRPO as well as ReAct-based agent variants.

The results show that the approach improves the performance of the search agents compared to GRPO, and is also on par or superior when compared to other agents.

**Strengths:**

- Well-motivated improvement: Model intermediate rewards for partially correct rollouts
- Improvement over GRPO visible in empirical evaluation
- Stabilization of training process increases efficiency

**Weaknesses:**

- Empirical gain rather low compared to GRPO

**Questions:**

- Did you experiment with decreasing alpha during training?
- Can this approach be transfered to other domains with less entities?

---

> ### Author Response · Authors · 2025-11-22
> **Response to Reviewer oC3b**
>
> We sincerely appreciate the effort and time you have dedicated to evaluating our work.
> > Question 1: Decreasing alpha during training.
>
> Thank you for this excellent suggestion. We additionally train the 30B model with an alpha linearly decaying from 0.3 to 0.0 in the first 60 steps (80 steps in total). The results are presented below and added to **Appendix F** of our modified paper.
> |alpha|GAIA|BrowseComp|BrowseComp-ZH|xbench-DS|
> |-----|----|----------|-------------|---------|
> |0.3|48.5|12.9|26.4|46.7|
> |decay|48.2|12.8|26.2|47.3|
>
> Comparing with a fixed 0.3, the results show no clear or consistent advantage for the decaying alpha strategy. This suggests our paper's choice of a simple, fixed alpha as a practical and effective setting. However, we believe your intuition that a dynamic alpha holds potential is sound. It's possible that applying the decaying schedule over longer training horizons could be more impactful, which is a promising direction for future research.
>
> > Question 2: Can this approach be transferred to other domains with less entities?
>
> It can be transferred to other domains as long as there is reliable evidence for the reasoning path (e.g., key theorems, lemmas, or intermediate equations for math problems). This would involve replacing the entity match rate in our reward function with a metric that quantifies the matching of the corresponding domain-specific evidence.

---

> ### Author Response · Authors · 2025-11-25
> **Request for Feedback**
>
> We have tried our best to address the issues raised in your review. Could you please let us know if there are any remaining concerns?

---

### Author Response · Authors · 2025-11-22
**Summary of Paper Revision**

We sincerely appreciate all reviewers for their insightful and constructive feedback. In response to these valuable comments, we have made the following revisions to our paper:

1. **Added an Experiment on Decaying Alpha**: We have included results of training with a decaying alpha schedule in **Appendix F** (Requested by Reviewer oC3b).

2. **Conducted a Systematic Failure Analysis**: We performed an extended case study to identify and categorize common failure modes, presenting the analysis in **Appendix E** (Suggested by Reviewer fNgM).

3. **Introduced a "Robust Matching" Strategy**: To address entity variations, we designed, analyzed, and tested a new "Robust Matching" strategy, with full details and results in **Appendix H** (Suggested by Reviewer fNgM).

4. **Extended Training Dynamics**: We extended the training process by 40 steps to show longer-term performance trends, with the new learning curves plotted in **Appendix G** (Requested by Reviewer 4vB2).

5. **Clarified Experimental Setups**: We have expanded on the experimental details in **Section 4.1** (e.g., lines 354-357 for data synthesis) to improve clarity (In response to feedback from Reviewer LHoD).

---

### Author Response · Authors · 2025-12-02
**Rebuttal Summary for AC**

Dear Program Chairs, Senior Area Chairs, and Area Chairs,

We sincerely thank you and all reviewers (oC3b, fNgM, 4vB2, LHoD) for your time and valuable feedback. During the rebuttal, we provided detailed responses and new experiments to **address every concern raised**.

Here is a summary of the key points:

1. `Reviewer fNgM` acknowledged that their concerns were resolved and has **explicitly recommended acceptance with full confidence**.

2. The main concerns from `Reviewer 4vB2` and `Reviewer LHoD` **stemmed from a misunderstanding of our experimental setup**. We have clarified these points in the revised paper and added substantial new results to directly resolve them. All revisions are marked in `blue` in the revised PDF for your convenience. Although they did not follow up, we are confident their initial critiques have now been thoroughly addressed.

We fully understand that this is an exceptionally busy period for ACs. We hope these positive outcomes of the rebuttal process will be fully considered in your final decision.

Thank you again for your time and for handling this challenging situation.

Sincerely,

Authors of Paper #9516

---

### Meta-Review · Area_Chair_tEPL · 2026-01-07

**Summary:**

This paper proposes a new novel entity-centric reward assignment process for web-search agents trained with GRPO. Since the synthetic data generation process in search agents can be grounded in Wikipedia entities, this data is already available to be used more directly. The authors extend GRPO by assinging partial rewards when some of the entities required to reach the final correct answer appear in the agent’s trajectory. Their empirical results find that E-GRPO outperforms GRPO and leads to more efficient trajectories.

All of the reviewer’s concerns are addressed appropriately, I recommend accepting this paper.

For the final version, I expect that the authors will include their comparison with the NL2SQL literature mentioned by R3 and tune down the description of their method.

**Reviewer Concerns:**

R1:
- Low empirical gain vs GRPO
	- Authors do not address this.
- Decreasing alpha during training
	- They try this and find no improvement over the paper’s fixed alpha setting.
- Can this work with fewer entities?
	- Additional intermediate concepts such as theorems, lemmas or intermediate equations would have to be used instead of entities.

R2: (Reviewer kept score of 6)
- Incorrect reasoning but correct entities
	- Figure 1 shows strong correlation between correct entities and correct reasoning
- Overclaim
	- The authors concede and will tone down the descriptions of the method
- Insufficient baselines for local experiments
	- Many baselines are trained on real world web-search so would be unfair to evaluate them in a local environment.
- Unexplained performance degradation on PopQA
	- PopQA is too simple and unambiguous to truly require the benefits usually attributed to E-GRPO
- Error analysis missing
	- Added an error taxonomy and some error examples to the paper.
- Entity matching robustness concerns
	- An experiment based on this concern was added to the paper.

R3:
- Incremental contribution (similar partial entity matching in RL is established in NL2SQL literature)
	- The authors concede that they are not inventing partial-match rewards but rather effectively implementing this within search agent RL training. They also distinguish their contribution from NL2SQL by noting that Reasoning-SQL measures the component-wise correctness of the outcome, not the reasoning path. This response explains the significance of their contribution clearly and contextualizes it well, I expect this to be added to the final paper.
- Training data differences with other methods?
	- They claim that it is impossible to compare fairly with other search agents (in terms of training data parity) since their methods are inseparable from their training data. However, they do provide explicit comparisons with standard GRPO on the same training data to support their main claim. This seems like a reasonable experimental approach.
- Is uniformly assigning entity matching rewards technically sound? Why not do turn or token based reward assignment?
	- Assigning entity-level rewards to the whole trajectory leaves most of GRPO unchanged and therefore keeps its stability. Assigning turn-level rewards would not work because some turns (such as planning moves) might not contain entities but are important. This again is a reasonable response.
- Are the methods converging or is the difference seen just an artifact of # of training steps?
	- The authors continue training past 80 steps to 120 and the E-GRPO gains continue.

R4:
- Outcome is expected
	- The authors argue that implementing and evaluating a reasonable idea is still important empirical work and I tend to agree.
- Multiple valid reasoning paths
	- They use the same answer as the first one in R2, Figure 1 shows strong correlation between correct entities and correct reasoning.
- Incremental contribution
	- The authors concede that their contribution is a novel reward design method and will change the descriptions of the method in the final version.
- Clarify the difference between PRM vs E-GRPO
	- PRMs require expensive step-by-step annotations while E-GRPO requires only gold entities.
- Robustness to synthetic data quality (in terms of entity extraction) AND generalization to real-world search questions
	- The synthetic data generated for training this work is grounded in Wikipedia entities and generalizes to web-search benchmarks like GAIA, BrowseComp and others. Since these are out-of-domains benchmarks for the synthetic data generated, they provide strong evidence for both the robustness to synthetic data and the generalizability to real-world search queries.

**Reviewer Scores:**

- oC3b 6 -> 6
	- Reviewer was most likely to keep a positive score with low certainty since their weaknesses were not serious and were addressed well.
- fNgM 6 -> 6
	- Reviewer kept positive score with highest certainty.
- 4vB2 4 -> 6
	- The response was strong and it should have convinced the reviewer.
- LHoD 4 -> 6
	- The response was strong and it should have convinced the reviewer.

---

### Decision · Program_Chairs · 2026-01-26

Accept (Poster)